# Boosts in brain signal variability track liberal shifts in decision bias

**Niels A Kloosterman[1,2]\*, Julian Q Kosciessa[1,2], Ulman Lindenberger[1,2], Johannes Jacobus Fahrenfort[3,4], Douglas D Garrett[1,2]\***

[1]Max Planck UCL Centre for Computational Psychiatry and Ageing Research, Berlin, Germany; [2]Center for Lifespan Psychology, Max Planck Institute for Human Development, Berlin, Germany; [3]Department of Experimental and Applied Psychology, Vrije Universiteit Amsterdam, Amsterdam, Netherlands; [4]Department of Psychology, University of Amsterdam, Amsterdam, Netherlands

**Abstract** Adopting particular decision biases allows organisms to tailor their choices to environmental demands. For example, a liberal response strategy pays off when target detection is crucial, whereas a conservative strategy is optimal for avoiding false alarms. Using conventional time-frequency analysis of human electroencephalographic (EEG) activity, we previously showed that bias setting entails adjustment of evidence accumulation in sensory regions (Kloosterman et al., 2019), but the presumed prefrontal signature of a conservative-to-liberal bias shift has remained elusive. Here, we show that a liberal bias shift is reflected in a more unconstrained neural regime (boosted entropy) in frontal regions that is suited to the detection of unpredictable events. Overall EEG variation, spectral power and event-related potentials could not explain this relationship, highlighting that moment-to-moment neural variability uniquely tracks bias shifts. Neural variability modulation through prefrontal cortex appears instrumental for permitting an organism to adapt its biases to environmental demands.

**\*For correspondence:**
kloosterman@mpib-berlin.mpg.de (NAK);
Garrett@mpib-berlin.mpg.de (DDG)

**Competing interests:** The authors declare that no competing interests exist.

## Introduction

We often reach decisions not only by objectively weighing different alternatives, but also by allowing subjective biases to influence our choices. Ideally, such biases should be under internal control, allowing us to flexibly adapt to changes in task context while performing a challenging task. Specifically, contexts which prioritize target detection benefit from a liberal response strategy, whereas a conservative strategy should be used at times when it is important to avoid errors of commission (e.g. false alarms). Adaptive shifts in decision bias are presumed to rely on prefrontal cortex (*Rahnev et al., 2016*), but despite growing interest (*Chen et al., 2015*; *Reckless et al., 2014*; *Windmann et al., 2002*), the spatio-temporal neural signature of such within-person bias shifts is currently unknown.

One candidate neural signature of decision bias shifts that has not been considered thus far is the variability of brain activity, as reflected in its moment-to-moment irregularity. Temporal neural variability is a prominent feature in all types of neural recordings (single-cell, local field potentials, EEG/MEG, fMRI) and has traditionally been considered noise that corrupts neural computation (*Dinstein et al., 2015*; *Faisal et al., 2008*). In contrast, heightened neural variability is increasingly proposed to support cognitive flexibility by allowing the brain to continuously explore its dynamical repertoire, helping it to quickly adapt to and process a novel stimulus (*Ghosh et al., 2008*; *Misić et al., 2010*). Indeed, a growing body of evidence suggests that neural variability can prove optimal for neural systems, allowing individuals to perform better, respond faster, and adapt quicker to their environment (*Garrett et al., 2015*, *Garrett et al., 2013a*, *Garrett et al., 2011*).

One tangible possibility is that cognitive flexibility emerges when a neural system avoids locking into a stereotypical, rhythmic pattern of activity, while instead continuously exploring its full dynamic range to better prepare for unpredictably occurring events. Consistent with this notion of exploration, influential attractor models of neural population activity (*Chaudhuri et al., 2019*; *Inagaki et al., 2019*; *Wimmer et al., 2014*) typically contain a 'noise' component that drives a dynamical system from attractor state to attractor state within a high-dimensional state space (*Deco et al., 2009*; *Deco and Romo, 2008*). This element of noise might indeed correspond to modulations of moment-to-moment neural variability during cognitive operations that can be empirically observed. Here, we perform a crucial test of the utility of temporal neural variability in the context of adaptive human decision making. In line with recent ideas that a high-fidelity, more variable neural encoding regime may be particularly required in more complex, non-deterministic situations (*Garrett et al., 2020*; *Marzen and DeDeo, 2017*; *Młynarski and Hermundstad, 2018*), we hypothesized that increased neural variability might underlie a state of higher receptiveness to and preparedness for events of interest that are not predictable in time, permitting the adoption of a more liberal bias toward confirming that such an event has indeed occurred.

Interestingly, improved cognitive function has also recently been linked to *reduced* neural variability, in line with the presumed corruptive role of noise for cognitive operations (*Faisal et al., 2008*). In particular, transient variability decreases after stimulus onset – called 'quenching' (*Churchland et al., 2010*) – have been proposed to reflect the settlement of an attractor into a stable state (*Churchland et al., 2010*; *Schurger et al., 2015*; *Wang, 2002*), with quenching reportedly being stronger during conscious perception relative to when a stimulus passes unseen (*Schurger et al., 2015*). Stronger quenching has also been reported in observers with higher perceptual sensitivity (*Arazi et al., 2017*), in line with a central assumption of signal detection theory (SDT) that internal noise is detrimental for sensitivity and thus should be suppressed (*Green and Swets, 1966*). To attend to this conceptual discrepancy, we further asked whether a quenching effect can also be observed in moment-to-moment variability, and if so, whether it reflects adaptive decision bias shifts and perceptual sensitivity.

We investigated these issues using previously published data from humans performing a continuous target detection task under two different decision bias manipulations, while non-invasively recording their electroencephalogram (EEG) (*Kloosterman et al., 2019*). Sixteen participants (three experimental sessions each) were asked to detect orientation-defined squares within a continuous stream of line textures of various orientations and report targets via a button press (*Figure 1A*). In alternating 9-min blocks of trials, we actively biased participants' perceptual decisions by instructing them either to report as many targets as possible (liberal condition), or to only report high-certainty targets (conservative condition). We played auditory feedback after errors and imposed monetary penalties to enforce instructions.

In our previous paper on these data, we reported within-participant evidence that decision bias in each condition separately is implemented by modulating the accumulation of sensory evidence in posterior brain regions through oscillatory EEG activity in the 8–12 Hz (alpha) and gamma (60–100 Hz) frequency ranges (*Kloosterman et al., 2019*). In no brain region, however, did we find a change-change relationship between participants' liberal–conservative shifts in decision bias and in spectral power, despite substantial available data per participant (on average 1733 trials) and considerable inter-individual differences in the bias shift. Reasoning that moment-to-moment variability of neural activity may instead better capture the adaptive bias shift from person to person, potentially revealing its hypothesized prefrontal signature, we here measured temporal variability in the EEG data using a novel extension of multi-scale entropy (MSE)(*Costa et al., 2002*). We then tested for a change-change relationship by correlating within-person liberal–conservative shifts in decision bias with those estimated via our modified MSE (mMSE) measure. We indeed found that those participants who shifted more toward a liberal bias (in line with task demands) also showed a stronger boost in mMSE. This relationship could not be explained by overall EEG signal variation, band-specific spectral power, and event-related potentials, highlighting the unique contribution of moment-to-moment neural variability to the bias shift. Finally, we show that interactions between spectral power and phase in low frequencies (1–3 Hz) may underlie the observed effects.

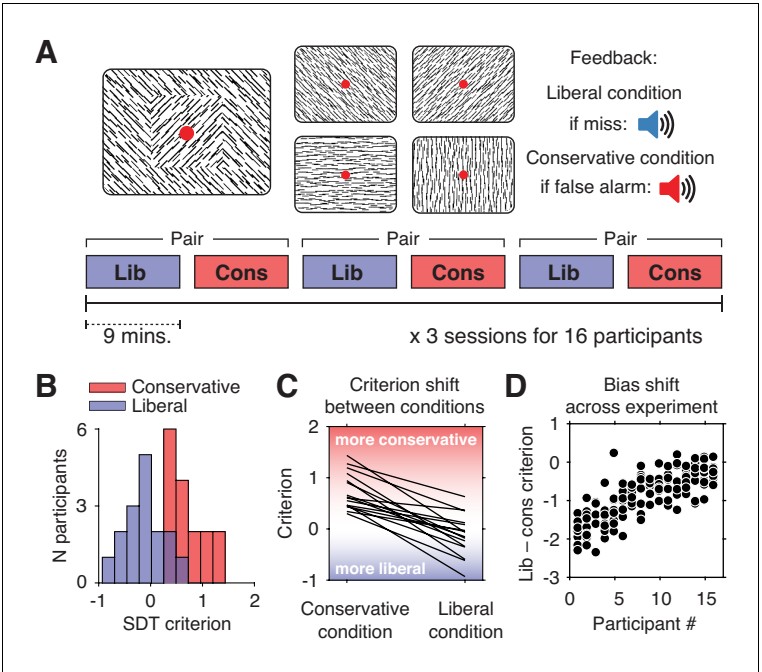

**Figure 1.** Experimental paradigm and behavioral results. (**A**) Top, target and non-target stimuli. Subjects detected targets (left panel) within a continuous stream of diagonal and cardinal line stimuli (middle panel), and reported targets via a button press. In different blocks of trials, subjects were instructed to actively avoid either target misses (liberal condition) or false alarms (conservative condition). Auditory feedback was played directly after the respective error in both conditions (right panel). Bottom, time course of an experimental session. The two conditions were alternatingly administered in blocks of nine minutes. In between blocks participants were informed about current task performance and received instructions for the next block. Subsequent liberal and conservative blocks were paired for within-participant analyses (see panel D, and *Figure 3C*). (**B**) Distributions of participants' criterion in both conditions. A positive criterion indicates a more conservative bias, whereas a negative criterion indicates a more liberal bias. (**C**) Lines indicating the criteria used by each participant in the two conditions, highlighting individual differences both in overall criterion (line intercepts), and in the size of the criterion shift between conditions (slopes). (**D**) Within-person bias shifts for liberal–conservative block pairs (see panel A, bottom). Participants were sorted based on average criterion shift before plotting.

The online version of this article includes the following figure supplement(s) for figure 1:

**Figure supplement 1.** Perceptual sensitivity and relationship between decision bias and sensitivity.

## Results

### Large individual differences in the extent of decision bias shift

Participants differentially adopted the intended decision biases in the respective conditions, as quantified by the criterion SDT measure of bias (*Green and Swets, 1966*). Subjects assumed a lower criterion (more liberal bias) when target detection was emphasized ($c = –0.13$, standard deviation (SD) 0.4) and adopted a higher criterion (more conservative bias) when instructed to avoid false alarms ($c = 0.73$, SD 0.36; liberal vs. conservative, p=0.001, two-sided permutation test, 1000 permutations) (*Figure 1B*). Participants varied substantially not only in the average criterion they used across the two conditions (range of $c = –0.24$ to 0.89), but also in the size of the criterion shift between conditions (range of $\Delta c = –1.54$ to $–0.23$). Highlighting the extent of individual differences, participant's biases in the two conditions were only weakly correlated (Spearman's rho = 0.24, p=0.36), as can be seen from the subjects' large variation in criterion intercept and slope between the two conditions (*Figure 1C*). Moreover, each participant's bias shift also fluctuated spontaneously over the course of the experiment, as indicated by variation in criterion differences between successive, nine-minute liberal and conservative blocks (participant-average SD 0.37, *Figure 1D*). Participants also varied widely in their ability to detect targets (range in SDT d´ 0.26 to 3.97), but achieved similar d´ in both bias

conditions (rho = 0.97, p<0.001, *Figure 1—figure supplement 1*). Moreover, the liberal–conservative bias shift was only weakly correlated with a shift in sensitivity across participants (rho = 0.44, p=0.09), indicating that the bias manipulation largely left perceptual sensitivity unaffected. In our previous paper on these data (*Kloosterman et al., 2019*), we also quantified decision bias in these data in terms of the 'drift bias' parameter of the drift diffusion model (*Ratcliff and McKoon, 2008*). We chose to focus on SDT criterion in the current paper due to its predominant use in the literature and its comparably simpler computation, while noting the substantial overlap between the two measures as indicated by their high correlation (rho = –0.89, as reported in *Kloosterman et al., 2019*). Taken together, we observed considerable variability in the magnitude of the decision bias shift as a result of our bias manipulation, both at the group level and within single individuals.

## Measuring neural variability with modified multi-scale entropy

We exploited the between- and within-participant variations in liberal–conservative criterion differences to test our hypothesis that a larger boost in brain signal variability should reflect a more liberal bias shift. To this end, we developed a novel algorithm based on multi-scale entropy (MSE) that directly quantifies the temporal irregularity of the EEG signal at shorter and longer timescales by counting how often temporal patterns in the signal reoccur during the signal's time course (*Costa et al., 2002*; *Figure 2A*, bottom). In general, signals that tend to repeat over time, such as neural oscillations, are assigned lower entropy, whereas more irregular, non-repeating signals yield higher entropy. Please see the Materials and methods section for a step-by-step description of the MSE computation in our EEG data.

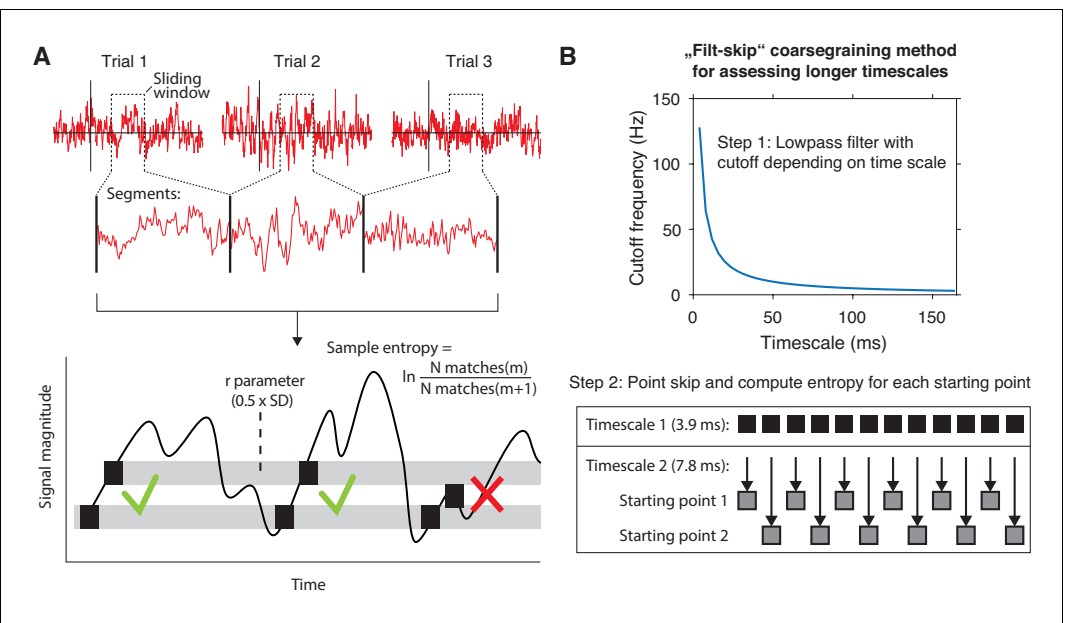

**Figure 2.** mMSE estimation procedure. (**A**) Discontinuous entropy computation procedure. Data segments of 0.5 s duration centered on a specific time point from each trial's onset (top row) are selected and concatenated (middle row). Entropy is then computed on this concatenated time series while excluding discontinuous segment borders by counting repeats of both m (here, m = 1 for illustration purposes) and m+1 (thus 2) sample patterns and taking the log ratio of the two pattern counts (bottom row). We used m = 2 in our actual analyses. The pattern similarity parameter r determines how lenient the algorithm is toward counting a pattern as a repeat by taking a proportion of the signal's standard deviation (SD), indicated by the width of the horizontal gray bars. The pattern counting procedure is repeated at each step of the sliding window, resulting in a time course of entropy estimates computed across trials. (**B**) 'Filt-skip' coarsegraining procedure used to estimate entropy on longer timescales, consisting of low-pass filtering followed by point-skipping. Filter cutoff frequency is determined by dividing the data sampling rate (here, 256 Hz i.e. 1 sample per 3.9 ms) by the index of the timescale of interest (top row). The signal is then coarsened by intermittently skipping samples (bottom row). In this example, every second sample is skipped at timescale 2, resulting in two different time courses depending on the starting point. Patterns are counted independently for both starting points and summed before computing entropy.

We developed time-resolved, modified MSE (mMSE), that differs from traditional MSE in two ways. First, slower timescales are usually assessed in conventional entropy analysis by 'coarsening' the data through averaging of data samples close to each other in time, and repeating the pattern counting operation (see *Figure 2A*). Although this method can remove faster dynamics from the data in a straightforward way, it is prone to aliasing artifacts and thereby possibly obscures genuine entropy effects in the data. Therefore, we instead coarsened the data by applying a Butterworth low-pass filter followed by skipping of data points (*Figure 2B*), thereby retaining better control over the frequencies present in the coarsened signal (*Semmlow, 2014*; *Valencia et al., 2009*). Second, conventional entropy analysis requires substantial continuous data (in the order of minutes) for robust estimation, which makes the standard method unsuitable for studying brief, transient cognitive operations such as perceptual decision making. To investigate entropy dynamics over time, we calculated entropy across discontinuous data segments aggregated across trials via a sliding window approach (*Grandy et al., 2016*; *Figure 2A*, top), allowing us to examine entropy fluctuations from moment to moment. Please see Materials and methods for details on the various analysis steps and our modifications of the MSE algorithm.

## Larger boosts in frontal entropy track more liberal decision bias shifts

We tested for a relationship between shifts in decision bias and neural variability from the conservative to the liberal conditions by Spearman-correlating joint modulations of mMSE and criterion across participants (averaged over the three sessions), for all electrodes, time points, and timescales. Strikingly, we found a negative cluster of correlations in mid- and left-frontal electrodes (p=0.015, cluster-corrected for multiple comparisons [*Maris and Oostenveld, 2007*]) indicating that participants who showed a larger bias shift from the conservative to the liberal condition were those who also exhibited a larger boost in frontal entropy (*Figure 3A*). The cluster ranged across timescales from ~20 to 164 ms, with most of the cluster located after trial initialization (solid vertical line in *Figure 3A*). To illustrate this correlation, we obtained a point estimate of mMSE per participant by averaging liberal–conservative mMSE within the significant cluster, and plotted the across-participant change-change correlation (rho = –0.87) with criterion in a scatter plot (*Figure 3B*). Since this correlation is bound to be significant due to averaging across significantly correlating time-timescale bins from the principal analysis, we consider it a descriptive statistic and refrain from reporting its p-value. We employ the participant-wise mMSE point estimates to examine the relationship with other neural measures (see next sections). In contrast to these correlational results, we found no significant clusters (main effect) when contrasting the two conditions. To provide an intuition of the mMSE values that fed into the correlation analysis, we plotted the subject-averaged mMSE values within the cluster separately for the two conditions (*Figure 3—figure supplement 1*). This indeed shows highly similar subject-average mMSE for the two conditions, highlighting the lack of a main effect of condition in our data. Taken together, we observed a strong change-change correlational link in frontal brain regions between liberal–conservative shifts in mMSE and decision bias, suggesting that participants with a stronger increase in temporal neural variability from to conservative to the liberal condition achieved a greater liberal bias shift.

## Entropy-bias correlations are also present within participants and in split data

Correlating brain and behavior across a relatively modest number of participants can be unreliable (*Yarkoni, 2009*), depending on the amount of data underlying each observation. Therefore, we next employed two complementary approaches to strengthen evidence for the observed link between shifts in neural variability and decision bias. We first asked whether mMSE and bias were also linked *within* participants across the nine liberal–conservative block pairs that each participant performed throughout the three sessions (see *Figure 1A*, bottom for task structure and *Figure 1D* for criterion shifts in single block pairs). Critically, we observed a negative repeated measures correlation (*Bakdash and Marusich, 2017*) between within-participant shifts in criterion and mMSE ($r_{rm}$ = –0.19, p=0.046, *Figure 3C*), providing convergent within-person evidence for a link between shifts in decision bias and neural variability. Second, we tested whether the observed across-participant correlation was present within two separate halves of the data after an arbitrary split based on odd and even trials. We found significant change-change correlations in both data halves, indicating reliable

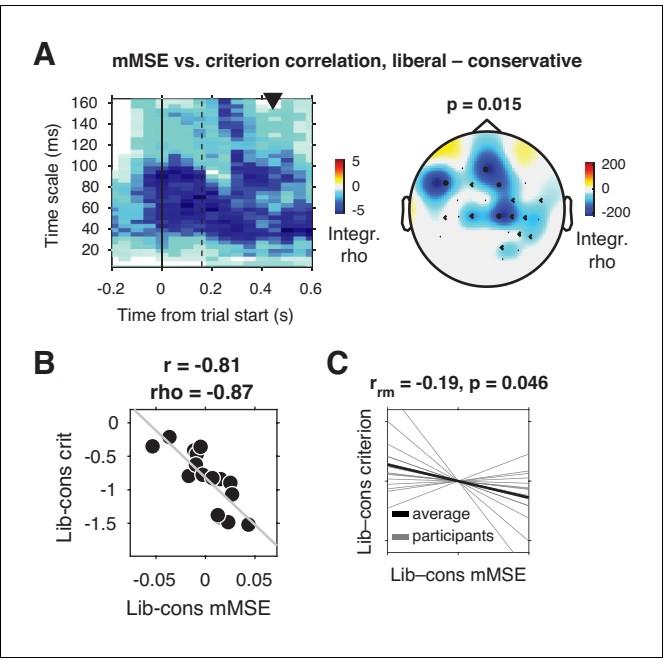

**Figure 3.** Change-change correlation between liberal–conservative shifts in mMSE and bias. (**A**) Significant negative electrode-time-timescale cluster observed via Spearman correlation between liberal–conservative mMSE and liberal–conservative SDT criterion, identified using a cluster-based permutation test across participants to control for multiple comparisons. Correlations outside the significant cluster are masked out. Left panel, time-timescale representation showing the correlation cluster integrated over the electrodes included in the cluster, indicated by the black dots in the topographical scalp map in the right panel. Dot size indicates how many time-timescale bins contribute to the cluster at each electrode. Color borders are imprecise due to interpolation used while plotting. The solid vertical line indicates the time of trial onset. The dotted vertical line indicates time of (non)target onset. Right panel, scalp map of mMSE integrated across time-timescale bins belonging to the cluster. p-Value above scalp map indicates statistical significance of the cluster. The black triangle indicates participants' median reaction time, averaged across participants and conditions. (**B**) Scatter plot of the correlation after averaging mMSE within time-timescale-electrode bins that are part of the three-dimensional cluster. Since the cluster was defined based on significant change-change correlations, averaging mMSE across the significant time-timescale-electrode bins before correlating represents no new information. Thus, the scatter plot serves only to illustrate the negative relationship identified in panel A. Both Pearson's r and Spearman's rho are indicated. We report no p-values since the bin selection procedure guarantees significance, and consider the correlation a descriptive statistic only. (**C**) Within-participant mMSE vs. criterion slopes across liberal–conservative block pairs completed across the experiment. $r_{rm}$, repeated measures correlation across all block pairs performed after centering each participant's shifts in mMSE and criterion around zero by removing the intercept. Gray lines, individual participant slopes fit across liberal–conservative mMSE vs criterion block pairs. Black line, slope averaged across participants.

The online version of this article includes the following source data and figure supplement(s) for figure 3:

**Source data 1.** This MATLAB file contains the data for *Figure 3A and B*.
**Source data 2.** This csv file contains the data for *Figure 3C*.
**Figure supplement 1.** Raw mMSE values averaged across subjects within the correlation cluster identified in *Figure 3A*.
**Figure supplement 2.** Correlations between liberal–conservative shifts in mMSE versus criterion in split data, and versus EEG signal SD and spectral power.
**Figure supplement 3.** Control analyses investigating signal SD and point averaging method.
**Figure supplement 4.** EEG spectral power normalized by subtracting the condition-average pre-trial baseline.
**Figure supplement 5.** Event-related responses in both conditions.

between-subject associations (odd, rho = –0.77, p=0.001; even, rho = –0.75, p=0.001)(*Figure 3—figure supplement 2A and 2B*). In contrast to the significant change-change correlation, we found no significant single-condition correlations between mMSE and criterion (conservative: rho = –0.12, p=0.66, liberal: rho = –0.21, p=0.43) and no significant difference in correlation strength (Δrho = –

0.09, p=0.7, non-parametric correlation difference test, 10.000 permutations). This indicates that the change-change correlation was not driven exclusively by one of the two conditions, but rather that their difference reveals the strong relationship observed in the present data.

## The entropy-bias relationship is not explained by total signal variation or spectral power

Next, we investigated whether the entropy-behavior correlation could alternatively be explained by total signal variation (quantified via the signal SD), or spectral power. Specifically, the variance structure of a signal can influence entropy estimates through the pattern similarity (r) parameter (width of gray bars in *Figure 2*), even when this parameter is recomputed for each timescale after coarsening, as we did (*Kosciessa et al., 2020*). In addition, E/MEG data is often quantified in terms of oscillatory spectral power in canonical delta (1–2 Hz), theta (3–7 Hz), alpha (8–12 Hz), beta (13–30 Hz) and gamma (60–100 Hz) bands, which might be able to explain the entropy results through a similar dependency. (See *Kloosterman et al., 2019* for detailed spectral analysis of the current dataset). Therefore, we tested whether the Δbias-Δentropy correlation could be explained by broadband signal SD and band-specific spectral power. To make the computation of spectral power and entropy as similar as possible, we used the same 0.5 s sliding window and 50 ms step size for spectral analysis (1 s window to allow delta power estimation, see Materials and methods), and selected spectral power within the same electrodes and time points in which the mMSE effect was indicated.

Strikingly, we found that the Δbias-Δentropy correlation remained strong and significant both when controlling for signal SD (partial rho = −0.82, p<0.0001), and even when controlling for all major power bands simultaneously (delta, theta, alpha, beta, gamma; partial rho = −0.75, p=0.005). See *Figure 3—figure supplement 2* for correlations between mMSE and various potentially confounding factors. We also found similar results when separately controlling for signal SD within each time-timescale bin while correlating modulations of mMSE and criterion in all electrodes, time points, and timescales (*Figure 3—figure supplement 3A*). Importantly, the results did depend on our modified entropy estimation method, since the frontal correlation cluster was smaller and non-significant when performing the Δbias-Δentropy correlation using conventional MSE combined with our novel sliding window approach (cluster p=0.37) (*Costa et al., 2002*; *Figure 3—figure supplement 3B*). In contrast to mMSE, spectral power was not linked to the bias shift. We found no significant clusters when correlating the liberal–conservative shifts in bias versus raw spectral power and versus percent signal change power modulation using either the condition-specific pre-stimulus baseline, or a condition-average baseline subtraction (*Figure 3—figure supplement 4*). Finally, statistically controlling for the participants' perceptual ability to detect targets, quantified as the liberal–conservative shift in SDT sensitivity measure d′ (*Green and Swets, 1966*) did not affect the relationship (partial rho = −0.88, p<0.0001), indicating that perceptual sensitivity could not explain our results. Taken together, neither overall signal variation, nor spectral power, nor perceptual sensitivity could account for the observed correlation between shifts in mMSE and decision bias, highlighting the unique ability of mMSE to capture these behavioral differences.

## Entropy-bias relationship is not explained by event-related potentials

Next, we investigated whether event-related potentials (ERPs), as a relatively simple and widely used EEG metric, could explain the observed link between shifts in criterion and entropy (*Luck et al., 2000*). We computed ERPs for the liberal and conservative conditions by averaging trial time courses and tested them against each other for all electrodes and time points using the cluster-based permutation test. We observed one significant positive cluster (p=0.001, cluster-corrected, indicating a stronger ERP for the liberal condition) over central and parietal electrodes, and one negative cluster (p=0.001) in midfrontal, central and parietal electrodes (*Figure 3—figure supplement 4*). The timing and topography of the positive ERP cluster closely corresponded to the centroparietal positivity (CPP) signal thought to reflect sensory evidence accumulation (*O'Connell et al., 2012*). This is in line with our previous report of increased evidence accumulation in the liberal condition (*Kloosterman et al., 2019*). Importantly, when repeating the change-change correlation between liberal–conservative ERPs and criterion as performed for mMSE, we found no significant clusters (lowest p-value positive cluster, p=0.69; negative cluster, p=0.23). Furthermore, we repeated the mMSE analysis after removing the ERP from the overall EEG activity by subtracting the event-related

potential (computed by averaging all trials within a condition, session, and participant) from each single trial. ERP subtraction from single trials is typically performed to remove stimulus-evoked activity and focus on ongoing or 'induced' neural activity (*Klimesch et al., 1998*). The bias-entropy correlation remained virtually unchanged after removing the ERP (rho = –0.90 with removal versus rho = –0.87 without removal). See *Mišić et al., 2010* for similar evidence of independence of MSE from ERPs. Taken together, these results suggest that liberal–conservative ERPs cannot account for the brain-behavior link observed between shifts in bias and entropy.

## Entropy-bias relationship possibly mediated by power-phase interactions in the delta range

Given that overall signal variation, spectral power, and ERPs were not able to explain our entropy findings, it remains an open question as to which aspect of the EEG signal underlies the observed link between entropy and decision bias shifts. Since we previously found stronger midfrontal oscillatory activity in these data in the delta/theta (2–6 Hz) frequency range (*Kloosterman et al., 2019*), we next examined the impact of systematically removing the lowest frequencies in the data on the strength of the observed brain-behavior relationship. To this end, we performed entropy analysis after applying a high-pass filter with 1, 2, or 3 Hz cutoff frequencies to the time-domain data (filter order of 4, trials mirror-padded to 4 s to allow robust estimation [*Cohen, 2014*]). Note that we applied a 0.5 Hz high-pass filter during data preprocessing to remove slow drift in all cases. To quantify the strength of the brain-behavior correlation at each filtering step, we averaged mMSE within the time-space-timescale cluster that showed the strong negative correlation in our principal (non-filtered) analysis (see *Figure 3B*) and plotted scatterplots of the correlation.

*Figure 4* shows the results of this analysis for non-filtered data (*Figure 4A*, copied from *Figure 3B*), as well as for 1, 2, and 3 Hz high-pass filters (*Figure 4B-D*). Interestingly, we found that the brain-behavior relationship progressively weakened with higher cutoff frequencies, such that the correlation was non-significant after applying a 3 Hz high-pass filter before entropy analysis. Whereas this finding suggests that these low frequencies contribute to our entropy effects, our control analysis in *Figure 3—figure supplement 2I* indicates that statistically controlling for 1–2 Hz (delta) power does not affect the brain-behavior relationship. One explanation for these seemingly incongruent results could be the different ways in which oscillatory phase is treated in these two analyses: whereas statistically controlling for delta power does not take delta phase into account, the high-pass filter removes both power and phase information from the signal before entropy is computed. Taken together, these analyses reveal that the lowest frequencies present in the data might play a role in the entropy-behavior relationship, possibly through non-linear interactions between spectral power and phase of these frequencies.

## Entropy quenching is not related to behavior

Finally, we tested whether variability 'quenching' was related to behavior in our data. Specifically, improved perceptual sensitivity has been linked to transient, post-stimulus decreases in neural

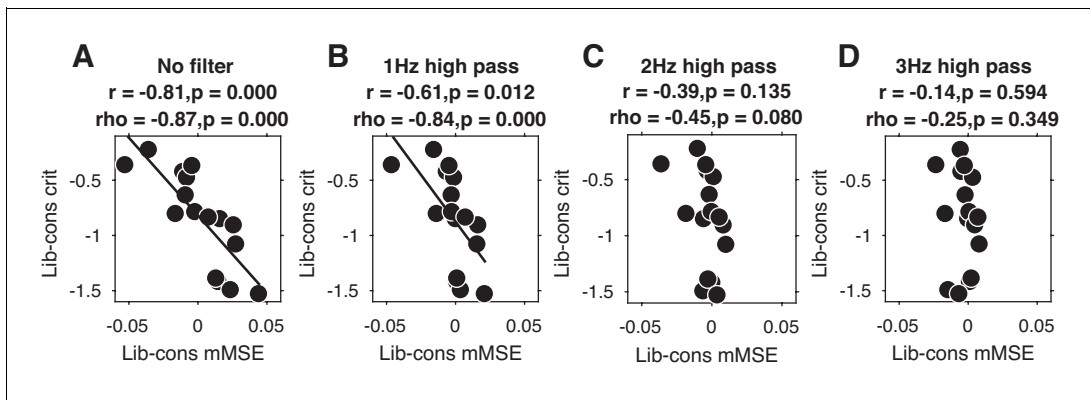

**Figure 4.** Liberal–conservative change-change correlation between mMSE and decision bias for non-filtered data (**A**) and after 1,2, and 3 Hz high-pass filtering (**B C** and **D**).

variability (*Arazi et al., 2017*; *Churchland et al., 2010*; *Schurger et al., 2015*). Quenching is directly predicted by attractor models of brain organization (*Wang, 2002*) and is consistent with the main principle of signal detection theory that suppression of neural noise enhances perception (*Green and Swets, 1966*). Quenching has also been reported in the human EEG over visual cortex in terms of a variance reduction across trials following stimulus onset (*Arazi et al., 2017*), although this type of quenching can be attributed to the well-known suppression of low-frequency (alpha and beta) spectral power following stimulus onset (*Daniel et al., 2019*). To our knowledge, only *Arazi et al., 2017* report an across-participant correlational link between perceptual sensitivity and variance quenching; however, in that study, this correlation could be explained by *elevated* absolute variability in the pre-stimulus period and not by brain activity in the post-stimulus period, suggesting that higher pre-stimulus variability was the more relevant factor for behavior. Nonetheless, we tested the link between entropy quenching and behavior in our data without any strong prior hypothesis.

To investigate this issue, we computed mMSE quenching by converting the raw mMSE values into percentage modulation from the pre-stimulus baseline and testing this modulation against zero. Besides a lateral occipital enhancement of mMSE modulation (*Figure 5A*) that could not be explained by spectral power modulation (*Figure 5B*), we also found a suppression of mMSE with a focal, mid-occipital topography, in line with quenching (*Figure 5C*). This focal topography was highly similar to that of the SSVEP evoked by the strong, visual stimulation frequency at 25 Hz (see Figure 3A of *Kloosterman et al., 2019*. In addition, spectral analysis of the ERP-subtracted data also revealed involvement of the subharmonic of this stimulation frequency at 12.5 Hz (data not shown). The highly periodic nature of this boosted SSVEP is bound to decrease the temporal irregularity of the signal, which could explain the observed mMSE suppression. In addition, the effect is strongest

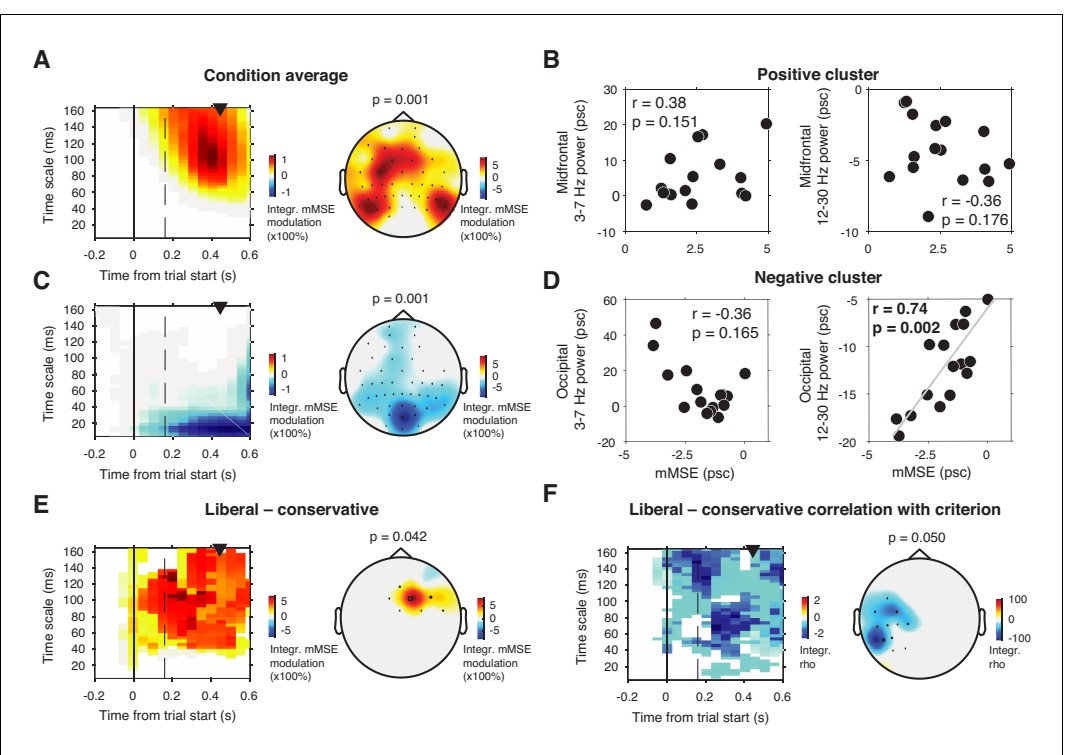

**Figure 5.** mMSE modulation with respect to pre-trial baseline. (**A**) Significant positive cluster observed in longer timescales after normalizing mMSE values to percent signal change (psc) units with respect to the pre-trial baseline (–0.2 to 0 s) and averaging across conditions. (**B**) Correlation between mMSE modulation in the positive cluster depicted in A and spectral power modulation in midfrontal electrodes. Left panel, 3–7 Hz; right panel, 12–30 Hz. (**C and D**) As B but for the posterior negative cluster. (**E**) Significant positive cluster observed in mid-frontal electrodes in the liberal–conservative contrast of mMSE modulation. (**F**) Significant cluster resulting from the correlation between liberal–conservative mMSE modulation with liberal–conservative SDT criterion. Conventions as in *Figure 3*.

in shorter time scales below 40 ms because of the progressive low-pass filter implemented for longer timescales in the coarse graining procedure, which removes these SSVEP-related frequencies from timescales slower than ca. 40 ms. mMSE quenching was indeed strongly positively correlated with low-frequency power encompassing 12.5 and 25 Hz (*Figure 5D*). Thus, the strongly periodic SSVEP boost after stimulus onset likely increased the temporal regularity of the EEG signal, which in turn suppressed post-stimulus entropy and manifested as quenching.

Contrasting transient mMSE percentage modulation between the two conditions, we found a significant positive cluster in midfrontal electrodes, indicating a stronger transient mMSE increase following trial onset in the liberal condition (*Figure 5E*). Furthermore, when change-change correlating liberal–conservative mMSE modulation and criterion, we observed a left-lateralized negative cluster in temporal electrodes, but no cluster in occipital electrodes (*Figure 5F*). Finally, we found no significant cluster when correlating liberal–conservative mMSE quenching with shifts in perceptual sensitivity (d′). Taken together, although we did find occipital, transient entropy quenching likely due to the strengthened SSVEP response, we found no convincing link between entropy quenching and behavior.

## Discussion

The ability to engage decision biases allows organisms to adapt their decisions to the context in which choices are made. Frontal cortex has previously been shown to be involved in adaptive bias shifts in humans (*Chen et al., 2015*; *Rahnev et al., 2016*; *Reckless et al., 2014*; *Windmann et al., 2002*) and monkeys (*Ferrera et al., 2009*), but its spatiotemporal neural signature has to date remained elusive. Here, we provide first evidence that greater bias shifts are typified in those subjects who exhibit greater shifts in frontal mMSE after stimulus onset, suggesting that mMSE provides a core signature of such adaptive behavioral shifts. Importantly, the relationship occurred independent of total brain signal variation, oscillatory neural dynamics, and ERPs. Moreover, it was observed at longer time scales, for which estimation was biased in a large amount of previous work (*Kosciessa et al., 2020*). Since the results were exclusively observed with principled extensions of the mMSE algorithm, our finding provides initial evidence for the unique value of brain signal irregularity at longer time scales.

The observed relationship between shifts in bias and neural variability in anterior brain regions complements our previous findings in the frequency domain that humans can intentionally control prestimulus 8–12 Hz (alpha) oscillatory power in posterior regions to adaptively bias decision making (*Kloosterman et al., 2019*). Notably, we previously observed increased oscillatory 2—6 Hz (theta) power in the liberal condition in the same midfrontal electrodes implicated here in the Δbias-Δ entropy correlation, but this theta power difference was not significantly correlated with the bias shift (rho = 0.23, p=0.39). This suggests that the bias shift may be reflected both in low-frequency spectral power and entropy in midfrontal regions, but that only entropy is linked to bias shift magnitude. One possible explanation for such a dissociation is that spectral power exclusively reflects the amplitude of oscillatory signal contributions while discarding their phase information. In contrast, entropy is sensitive to both variations in the magnitude as well as the phase of signal fluctuations. This notion is also in line with our finding that low-frequency spectral power is insufficient to explain our observed brain-behavior relationship, while the presence of these frequencies (including narrow-band non-linear phenomena such as phase resets or temporal dependencies in the amplitude of signals [*Linkenkaer-Hansen et al., 2001*]) during entropy estimation is sufficient and necessary for the relationship to emerge (*Figure 4*). Moreover, whereas spectral analysis strictly assumes a sinusoidal waveform of EEG signal fluctuations (*Cole and Voytek, 2017*; *Jones, 2016*), entropy analysis is agnostic to the shape of the waveforms present in the signal. Entropy thus provides a more unrestricted description of moment-to-moment fluctuations in neural activity that is highly predictive of decision bias shifts across participants in our data. Our results suggest that entropy taps into information in the EEG signal that is not available in ERPs and spectral power – the most popular analysis methods used in the field. Intriguingly, this suggests that many previous E/MEG studies analyzing ERPs and/or spectral power might have structurally overlooked a crucial property of the EEG signal that is in fact strongly linked to behavior. It could thus be that many interesting brain-behavior links are still hidden in existing EEG datasets, which can now be uncovered using mMSE.

Despite the consistent liberal decision bias shift that participants exhibited between the two conditions and the strong change-change correlation between entropy and behavior, mMSE was not significantly higher on average in the liberal condition. This is in contrast to the raw alpha and theta power differences that we reported previously, which did show significant condition differences (see *Kloosterman et al., 2019*). Strikingly, we show here that the shift in mMSE did correlate with the liberal–conservative bias shift, whereas the shifts in alpha and theta did not. This indicates that inter-individual differences in mMSE may be sensitive to behavioral adjustments even in the absence of a group-wise shift. A possible explanation for such a dissociation is that a main effect and a correlation address somewhat divergent research questions that may conflict with each other. On the one hand, a main effect tends to occur when subjects respond similarly to an experimental condition, which typically requires that individual differences be relatively small. On the other hand, chances of detecting a correlation with behavior typically increase when individual differences are larger, (e.g. *Lindenberger et al., 2008*). Thus, the common presumption that a main effect of condition is a prerequisite for detecting behavioral effects is unfounded in our view. Therefore, we did not a priori hypothesize a main effect of condition in the liberal–conservative contrast, but rather focused our hypotheses on inter-individual adjustments in mMSE that tracked the magnitude of the individual bias shift. We indeed observed a significantly stronger mMSE transient increase (main effect) after trial onset in the liberal condition once the data were baseline corrected (as is common in time-frequency EEG analysis), but the change-change correlation with behavior was weaker using baseline corrected mMSE (*Figure 5F*).

In apparent contrast to the view that neural variability facilitates cognition, previous work has suggested that a temporary stabilization of neural activity after stimulus onset (quenching, quantified as a transient suppression of time-domain variance) is beneficial for perception (*Arazi et al., 2017*; *Schurger et al., 2015*). We also observed a suppression in baseline-corrected mMSE, likely due to increased regularity of the time-domain signal due to the boosted power at the SSVEP frequency (*Kloosterman et al., 2019*). Previous work has linked variance quenching to post-stimulus suppression of rhythmic low-frequency (alpha and beta) power (*Daniel et al., 2019*). Future work could investigate whether entropy indeed increases after suppression of these temporally regular signals. Crucially, however – and divergent from our finding that boosting variability is coupled to an adaptive bias shift – we found no evidence for a change-change relationship between entropy quenching and decision bias or perceptual sensitivity. Since the relations between quenching observed in neural spiking (*Churchland et al., 2010*), trial-by-trial variance of E/MEG (*Arazi et al., 2017*), and mMSE are currently unclear, further investigation on this issue is needed (*Garrett et al., 2013b*). Future studies could also explore how neural variability quenching and boosting in different timescales are related to various aspects of decision making such as perceptual sensitivity and different kinds of biases (*Fleming et al., 2010*; *Talluri et al., 2018*; *Urai et al., 2019*), as well as to confidence and metacognitive processes (*Fleming and Dolan, 2012*; *Yeung and Summerfield, 2012*). Furthermore, individual decision bias has also been linked to the magnitude of transient dilations of the pupil (*de Gee et al., 2017*, *de Gee et al., 2014*) and to entropy of EEG (*Waschke et al., 2019*), suggesting that pupil-linked neuromodulation (*Joshi and Gold, 2020*) could be related to decision bias through adjustments to moment-to-moment neural variability. Further investigation of such relationships could yield fruitful insights about the neurochemical mechanisms underlying associations between neural variability and higher order cognitive function (*Alavash et al., 2018*; *Garrett et al., 2015*).

Our findings may have important implications for dynamical attractor models of neural population activity, which have become increasingly influential in recent years (*Chaudhuri et al., 2019*; *Inagaki et al., 2019*; *Wimmer et al., 2014*). Attractor models cast cognitive outcomes (e.g. decisions, perceptual experiences, or retention of an item in working memory) as low-dimensional, stable states ('attractors') within a high-dimensional energy landscape (*Deco et al., 2009*; *Deco and Romo, 2008*). These models typically contain a noise component that enables probabilistic exploration of the energy landscape, increasing chances of attraction to the most optimal state (e.g. the correct decision) and limiting the likelihood of settling too rigidly into any particular state. In multi-stable visual perception, for example, noise is thought to underlie the spontaneous, flexible switching between perceptual states reported by observers while viewing bi-stable visual illusions (*Kloosterman et al., 2015*; *Moreno-Bote et al., 2007*). Our results suggest that an element of noise facilitating cognitive flexibility might translate into modulations of moment-to-moment neural

variability that can be measured in cortical population activity. Future dynamical attractor modeling work could investigate exactly which characteristics of this noise component underlie effective exploration of the state space within these models, for example by modulating noise amplitude as well as the relative contribution of different noise frequencies (noise color). Modeling insights could then guide the search for signatures of noise supporting cognitive operations in moment-to-moment neural variability.

Our results suggest that dynamic adjustment of neural variability in frontal regions is related to adaptive behavior. Based on our findings, we speculate that heightened frontal entropy results from a more dynamic, irregular neural regime that enables an individual to be more prepared to process and act upon uncertain, yet task-relevant information. We believe that quantifying shifts in neural entropy could help elucidate the mechanisms allowing organisms to adapt to their environment and ultimately increase their chances of survival.

# Materials and methods

**Key resources table**

| Reagent type (species) or resource | Designation | Source or reference | Identifiers | Additional information |
|---|---|---|---|---|
| Biological sample (Humans) | Participants | *Kloosterman et al., 2019* | https://doi.org/10.7554/elife.37321 | See Subjects section in Materials and methods |
| Software, algorithm | MATLAB | Mathworks | MATLAB_R2016b, RRID:SCR_001622 | |
| Software, algorithm | Presentation | NeuroBS | Presentation_v9.9, RRID:SCR_002521 | |
| Software, algorithm | Statistical Analysis | R | R version 4.0.1, RRID:SCR_001905 | |
| Software, algorithm | Custom analysis code | *Kloosterman et al., 2019* | https://github.com/kloosterman/critEEG | |
| Software, algorithm | Custom analysis code | *Kloosterman et al., 2019* | https://github.com/kloosterman/critEEGentropy | |
| Software, algorithm | Custom analysis code | *Kloosterman, 2020* | https://github.com/LNDG/mMSE/ | FieldTrip-compatible toolbox |
| Other | EEG data experimental task | *Kloosterman et al., 2019* | https://doi.org/10.6084/m9.figshare.6142940 | |

We report a novel analysis of a previously published dataset involving a target detection task during two different decision bias manipulations (*Kloosterman et al., 2019*).

## Subjects

Sixteen participants (eight females, mean age 24.1 years,±1.64) took part in the experiment, either for financial compensation (EUR 10,– per hour) or in partial fulfillment of first year psychology course requirements. Each participant completed three experimental sessions on different days, each session lasting ca. 2 hr, including preparation and breaks. One participant completed only two sessions, yielding a total number of sessions across subjects of 47. Due to technical issues, for one session only data for the liberal condition was available. One participant was an author. All participants had normal or corrected-to-normal vision and were right handed. Participants provided written informed consent before the start of the experiment. All procedures were approved by the ethics committee of the University of Amsterdam.

## Stimuli

Stimuli consisted of a continuous semi-random rapid serial visual presentation (rsvp) of full screen texture patterns. The texture patterns consisted of line elements approx. 0.07° thick and 0.4° long in visual angle. Each texture in the rsvp was presented for 40 ms (i.e. stimulation frequency 25 Hz) and was oriented in one of four possible directions: 0°, 45°, 90° or 135°. Participants were instructed to fixate a red dot in the center of the screen. At random inter trial intervals (ITI's) sampled from a uniform

distribution (ITI range 0.3–2.2 s), the rsvp contained a fixed sequence of 25 texture patterns, which in total lasted one second. This fixed sequence consisted of four stimuli preceding a (non-)target stimulus (orientations of 45˚, 90˚, 0˚, 90˚, respectively) and twenty stimuli following the (non)-target (orientations of 0˚, 90˚, 0˚, 90˚, 0˚, 45˚, 0˚, 135˚, 90˚, 45˚, 0˚, 135˚, 0˚, 45˚, 90˚, 45˚, 90˚, 135˚, 0˚, 135˚, respectively) (see *Figure 1A*). The fifth texture pattern within the sequence (occurring from 0.16 s after sequence onset) was either a target or a nontarget stimulus. Nontargets consisted of either a 45˚ or a 135˚ homogenous texture, whereas targets contained a central orientation-defined square of 2.42˚ visual angle, thereby consisting of both a 45˚ and a 135˚ texture. 50% of all targets consisted of a 45˚ square and 50% of a 135˚ square. Of all trials, 75% contained a target and 25% a nontarget. Target and nontarget trials were presented in random order. To avoid specific influences on target stimulus visibility due to presentation of similarly or orthogonally oriented texture patterns temporally close in the cascade, no 45˚ and 135˚ oriented stimuli were presented directly before or after presentation of the target stimulus. All stimuli had an isoluminance of 72.2 cd/m$^2$. Stimuli were created using MATLAB (The Mathworks, Inc, Natick, MA) and presented using Presentation version 9.9 (Neurobehavioral systems, Inc, Albany, CA).

## Experimental design

The participants' task was to detect and actively report targets by pressing a button using their right hand. Targets occasionally went unreported, presumably due to constant forward and backward masking by the continuous cascade of stimuli and unpredictability of target timing (*Fahrenfort et al., 2007*). The onset of the fixed order of texture patterns preceding and following (non-)target stimuli was neither signaled nor apparent. At the beginning of the experiment, participants were informed they could earn a total bonus of EUR 30, -, on top of their regular pay of EUR 10, - per hour or course credit. In two separate conditions within each session of testing, we encouraged participants to use either a conservative or a liberal bias for reporting targets using both aversive sounds as well as reducing their bonus after errors. In the conservative condition, participants were instructed to only press the button when they were relatively sure they had seen the target. The instruction on screen before block onset read as follows: 'Try to detect as many targets as possible. Only press when you are relatively sure you just saw a target.' To maximize effectiveness of this instruction, participants were told the bonus would be diminished by 10 cents after a false alarm. During the experiment, a loud aversive sound was played after a false alarm to inform the participant about an error. During the liberal condition, participants were instructed to miss as few targets as possible. The instruction on screen before block onset read as follows: 'Try to detect as many targets as possible. If you sometimes press when there was nothing this is not so bad.' In this condition, the loud aversive sound was played twice in close succession whenever they failed to report a target, and three cents were subsequently deducted from their bonus. The difference in auditory feedback between both conditions was included to inform the participant about the type of error (miss or false alarm), to facilitate the desired bias in both conditions. After every block, the participant's score (number of missed targets in the liberal condition and number of false alarms in the conservative condition) was displayed on the screen, as well as the remainder of the bonus. After completing the last session of the experiment, every participant was paid the full bonus as required by the ethical committee.

Participants performed six blocks per session lasting ca. 9 min each. During a block, participants continuously monitored the screen and were free to respond by button press whenever they thought they saw a target. Each block contained 240 trials, of which 180 target and 60 nontarget trials. The task instruction was presented on the screen before the block started. The condition of the first block of a session was counterbalanced across participants. Prior to EEG recording in the first session, participants performed a 10 min practice run of both conditions, in which visual feedback directly after a miss (liberal condition) or false alarm (conservative) informed participants about their mistake, allowing them to adjust their decision bias accordingly. There were short breaks between blocks, in which participants indicated when they were ready to begin the next block.

## Behavioral analysis

We defined decision bias as the criterion measure from SDT (*Green and Swets, 1966*). We calculated the criterion $c$ across the trials in each condition as follows:

$$c = -\frac{1}{2}\left[Z(\textit{Hit-rate}) + Z(\textit{FA-rate})\right]$$

where hit-rate is the proportion target-present responses of all target-present trials, false alarm (FA)-rate is the proportion target-present responses of all target-absent trials, and Z(...) is the inverse standard normal distribution. Furthermore, we calculated perceptual sensitivity using the SDT measure d′:

$$d' = Z(\textit{Hit-rate}) - Z(\textit{FA-rate})$$

## EEG recording

Continuous EEG data were recorded at 256 Hz using a 48-channel BioSemi Active-Two system (Bio-Semi, Amsterdam, the Netherlands), connected to a standard EEG cap according to the international 10–20 system. Electrooculography (EOG) was recorded using two electrodes at the outer canthi of the left and right eyes and two electrodes placed above and below the right eye. Horizontal and vertical EOG electrodes were referenced against each other, two for horizontal and two for vertical eye movements (blinks). We used the FieldTrip toolbox (*Oostenveld et al., 2011*) and custom software in MATLAB R2016b (The Mathworks Inc, Natick, MA; RRID:SCR_001622) to process the data. Data were re-referenced to the average voltage of two electrodes attached to the earlobes. We applied a Butterworth high-pass filter (fourth order, cutoff 0.5 Hz) to remove slow drifts from the data.

## Trial extraction

We extracted trials of variable duration from 1 s before target sequence onset until 1.25 after button press for trials that included a button press (hits and false alarms), and until 1.25 s after stimulus onset for trials without a button press (misses and correct rejects). The following constraints were used to classify (non-)targets as detected (hits and false alarms), while avoiding the occurrence of button presses in close succession to target reports and button presses occurring outside of trials: 1) A trial was marked as detected if a response occurred within 0.84 s after target onset; 2) when the onset of the next target stimulus sequence started before trial end, the trial was terminated at the next trial's onset; 3) when a button press occurred in the 1.5 s before trial onset, the trial was extracted from 1.5 s after this button press; 4) when a button press occurred between 0.5 s before until 0.2 s after sequence onset, the trial was discarded. After trial extraction, the mean of every channel was removed per trial.

## Artifact rejection

Trials containing muscle artifacts were rejected from further analysis using a standard semi-automatic preprocessing method in Fieldtrip. This procedure consists of bandpass-filtering the trials of a condition block in the 110–125 Hz frequency range, which typically contains most of the muscle artifact activity, followed by a Z-transformation. Trials exceeding a threshold Z-score were removed completely from analysis. We used as the threshold the absolute value of the minimum Z-score within the block, + 1. To remove eye blink artifacts from the time courses, the EEG data from a complete session were transformed using independent component analysis (ICA), and components due to blinks (typically one or two) were removed from the data. In addition, to remove microsaccade-related artifacts we included two virtual channels in the ICA based on channels Fp1 and Fp2, which included transient spike potentials as identified using the saccadic artefact detection algorithm from *Hassler et al., 2011*. This yielded a total number of channels submitted to ICA of 48 + 2 = 50. The two components loading high on these virtual electrodes (typically with a frontal topography) were also removed. Blinks and eye movements were then semi-automatically detected from the horizontal and vertical EOG (frequency range 1–15 Hz; z-value cut-off four for vertical; six for horizontal) and trials containing eye artefacts within 0.1 s around target onset were discarded. This step was done to remove trials in which the target was not seen because the eyes were closed. Finally, trials exceeding a threshold voltage range of 200 mV were discarded. To attenuate volume conduction effects and suppress any remaining microsaccade-related activity, the scalp current density (SCD) was computed using the second-order derivative (the surface Laplacian) of the EEG potential distribution (*Perrin et al., 1989*).

## ERP removal

In a control analysis, we removed stimulus-evoked EEG activity related to external events by computing the event-related potential (ERP) and subtracting the ERP from each single trial prior to entropy or spectral analysis. This was done to focus on ongoing (termed 'induced', [*Klimesch et al., 1998*]) activity. To eliminate differences in evoked responses between sessions and conditions, we performed this procedure separately for ERPs computed in each condition, session, and participant.

## Entropy computation

We measured temporal neural variability in the EEG using a form of multiscale entropy (MSE) (*Costa et al., 2002*), which we modified in several ways. MSE characterizes signal irregularity at multiple time scales by estimating sample entropy (SampEn) of a signal's time series at various sampling rates. The estimation of SampEn involves counting how often specific temporal patterns reoccur over time, and effectively measures how unpredictably the signal develops from moment to moment. At a given time scale, the estimation of SampEn consists of the following steps:

1. A to-be-counted temporal 'template' pattern consisting of m samples is selected, starting at the beginning of the time series.
2. The data is discretized to allow comparing patterns of samples rather than exact sample values (which are rarely exactly equal in physiological timeseries). A boundary parameter $r$ is used to determine whether other patterns in the time series match the template. $r$ denotes the proportion of the time series standard deviation (SD, see also *Figure 2A*), within which a pattern is a match, as follows:

$$\text{Boundary parameter} = r \times SD \quad (1)$$

3. Template pattern repeats throughout the time series are counted, yielding pattern count $N(m)$.
4. Steps 1 to 3 are repeated for patterns consisting of $m + 1$ samples, yielding pattern count $N(m + 1)$.
5. Steps 1 to 4 are iterated to assess each temporal pattern as a template once. Counts for each template pattern are then summed, yielding total counts across templates for $N(m)$ and $N(m + 1)$.
6. Finally, SampEn is computed as the logarithm of the ratio of the counts for $m$ and $m + 1$:

$$\text{SampEn} = \ln \frac{N(m)}{N(m + 1)} \quad (2)$$

Thus, SampEn estimates the proportion of similar sequences of $m$ samples that are still similar when the next sample, that is $m + 1$, is added to the sequence. Here, we use $m = 2$ and $r = 0.5$, as typically done in neurophysiological settings (*Courtiol et al., 2016*; *Grandy et al., 2016*; *Richman and Moorman, 2000*).

We have implemented three modifications of regular MSE that we outline in the next sections. We refer to our own measure as modified MSE (mMSE) throughout the manuscript.

## MSE modification #1: multi-scale implementation through filtering and point skipping

In multiscale entropy, the computation of SampEn is repeated for multiple time scales after progressively lowering the time series sampling rate by a process called 'coarsening' (*Costa et al., 2002*). By default, SampEn quantifies entropy at the time scale that corresponds to the sampling rate of the time series, which is typically in the order of milliseconds or lower in (non-downsampled) neurophysiological data. To enable estimation of entropy at longer time scales, the time series is typically coarsening by averaging groups of adjacent samples ('point averaging') and repeating the entropy computation (*Costa et al., 2002*). However, despite being straightforward, this method is suboptimal for eliminating short temporal scales from the time series. Point averaging is equivalent to low-pass filtering using a finite-impulse response filter, which does not effectively eliminate high frequencies (*Semmlow, 2014*; *Valencia et al., 2009*). For this reason, we used an improved coarsening procedure involving replacement of the multi-point average by a low-pass Butterworth filter, which has

a well-defined frequency cutoff and precludes aliasing (*Valencia et al., 2009*; *Figure 2B*, top). The filter cutoff frequency *CutoffFreq* is determined as:

$$CutoffFreq = NyquistFreq \times \frac{1}{\text{scale number}} \qquad (3)$$

where *NyquistFreq* is the highest estimable frequency given the signal's sampling rate. This filtering ensures that an increasingly larger portion of the higher frequencies is removed for slower time scales. Note that low-pass filtering affects the temporal structure of the time-domain signal, which could hamper the interpretation of the EEG dynamics due to smearing of responses (*Vanrullen, 2011*). This issue is largely mitigated, however, due to the liberal–conservative subtraction that we perform before correlating with behavior, since this issue presumably affects both conditions similarly. Low-pass filtering is followed by a point-skipping procedure to reduce the sampling rate of the time series (*Figure 2B*, bottom). Since point-skipping omits increasingly large portions of the filtered time series depending on the starting point of the point-skipping procedure, we counted patterns separately for each starting point within a scale (see section Entropy computation above), summed their counts for $N(m)$ and $N(m+1)$ and computed entropy as described above.

## MSE modification #2: Pattern similarity recomputed at each time scale

By increasingly smoothing the time series, coarse-graining affects not only the signal's entropy, but also its overall variation, as reflected in the decreasing standard deviation as a function of time scale (*Nikulin and Brismar, 2004*). In the original implementation of the MSE calculation, the similarity parameter r was set as a proportion of the original (scale 1) time series' standard deviation and applied to all the scales (*Costa et al., 2002*). Because of the decreasing variation in the time series due to coarse graining, the similarity parameter therefore becomes increasingly tolerant at slower time scales, resulting in more similar patterns and decreased entropy. This decreasing entropy can be attributed both to changes in signal complexity, but also in overall variation (*Kosciessa et al., 2020*; *Nikulin and Brismar, 2004*). To overcome this limitation, we recomputed the similarity parameter for each scale, thereby normalizing mMSE with respect to changes in overall time series variation at each scale.

## MSE modification #3: Time-resolved computation

An important limitation of MSE is the need for substantial continuous data for robust estimation. Heuristically, the recommended number of successive data points for estimation at each scale is 100 (minimum) to 900 (preferred) points using typical MSE parameter settings (*Grandy et al., 2016*). This limitation precludes the application of MSE to neuroimaging data recorded during cognitive processes that unfold over brief periods of time, such as perceptual decisions. *Grandy et al., 2016* has shown that the pattern counting process can be extended to discontinuous data segments that are concatenated across time, as long as the counting of artificial patterns across segment borders is avoided (as these patterns are a product of the concatenation and do not occur in the data itself). We applied the mMSE computation across discontinuous segments of 0.5 s duration (window size). To track the evolution of mMSE over the trial, we slid this window across the trials in steps of 50 ms from −0.2 s until 0.6 s, each time recomputing mMSE across segments taken from the time window in each trial.

Given our segments of 0.5 s window length sampled at 256 Hz, we computed mMSE for scales 1 (129 samples within the window) until 42 (three or four samples within the window, depending on the starting point). Note that using a pattern parameter $m = 2$, a minimum of three samples within a segment is required to estimate entropy across the segments of continuous data, yielding a maximum possible scale of 42. In line with the MSE literature (*Courtiol et al., 2016*), we converted the time scale units to milliseconds by taking the duration between adjacent data points after each coarsegraining step. For example, time scale 1 corresponds to 1000 ms / 256 Hz = 3.9 ms, and time scale 42 to 1000 / (256/42) = 164 ms.

## Spectral analysis

We used a sliding window Fourier transform; step size, 50 ms; window size, 500 ms; frequency resolution, 2 Hz) to calculate time-frequency representations (spectrograms) of the EEG power for each

electrode and each trial. We used a single Hann taper for the frequency range of 3–35 Hz (spectral smoothing, 4.5 Hz, bin size, 1 Hz) and the multitaper technique for the 36–100 Hz frequency range (spectral smoothing, 8 Hz; bin size, 2 Hz; five tapers)(*Mitra and Bokil, 2007*). See *Kloosterman et al., 2019* for similar settings. Finally, to investigate spectral power between 1 and 3 Hz (delta band), we performed an additional time-frequency analysis with a window size of 1 s (i.e. frequency resolution 1 Hz) without spectral smoothing (bin size 0.5 Hz). Spectrograms were aligned to the onset of the stimulus sequence containing the (non)target. Power modulations during the trials were quantified as the percentage of power change at a given time point and frequency bin, relative to a baseline power value for each frequency bin. We used as a baseline the mean EEG power in the interval 0.4 to 0 s before trial onset, computed separately for each condition. If this interval was not completely present in the trial due to preceding events (see Trial extraction), this period was shortened accordingly. We normalized the data by subtracting the baseline from each time-frequency bin and dividing this difference by the baseline (x 100%). In an additional analysis, we performed a baseline correction by subtracting the condition-averaged pre-stimulus power, without converting into percent signal change.

### Statistical significance testing of mMSE and spectral power and correlations across space, time, and timescales/frequencies

To determine clusters of significant modulation with respect to the pre-stimulus baseline without any a priori selection, we ran statistics across space-time-frequency bins using paired t-tests across subjects performed at each bin. Single bins were subsequently thresholded at $p<0.05$ and clusters of contiguous time-space-frequency bins were determined. For the correlation versions of this analysis, we correlated the brain measure at each bin with the criterion and converted the r-values to a t-statistic using the Fisher-transformation (*Fisher, 1915*). We used a cluster-based procedure (*Maris and Oostenveld, 2007*) to correct for multiple comparisons using a cluster-formation alpha of $p<0.05$ and a cluster-corrected alpha of $p=0.05$, two-tailed (10.000 permutations). For visualization purposes, we integrated (using MATLAB's trapz function) power or entropy values in the time-frequency/entropy representations (TFR/TTR) across the highlighted electrodes in the topographies. For the topographical scalp maps, modulation was integrated across the saturated time-frequency bins in the TFRs/TTRs. See *Kloosterman et al., 2019* for a similar procedure in the time-frequency domain.

### High-pass filtering analysis

To examine the effect of systematically removing lower frequencies from the data before computing mMSE, we high-pass filtered the data using 1, 2 and 3 Hz high-pass filters (filter order of 4). We mirror-padded trials to 4 s to allow robust estimation (*Cohen, 2014*). After high-pass filtering, we performed mMSE analysis as reported above.

### Correlation analysis

We used both Pearson correlation and robust Spearman correlation across participants to test the relationships between the behavioral variables as well as with the EEG entropy and power (modulation). To test whether behavior and EEG activity were linked within participants, we used repeated measures correlation using the *rmcorr* package in R (*R Development Core Team, 2020*). Repeated measures correlation determines the common within-individual association for paired measures assessed on two or more occasions for multiple individuals by controlling for the specific range in which individuals' measurements operate, and correcting the correlation degrees of freedom for non-independence of repeated measurements obtained from each individual (*Bakdash and Maru-sich, 2017*; *Bland and Altman, 1995*). To test whether spectral power could account for the observed correlation between criterion and mMSE, we used partial Spearman and Pearson correlation controlling for other variables. To test whether the mMSE-bias correlation was stronger in any of the two conditions, we used a non-parametric correlation difference test. Specifically, data was shuffled 10,000 times within each correlation data pair, each time taking the difference between correlations to generate a distribution of correlations differences under the null hypothesis. Finally, the r difference of the actual correlations was compared to this distribution to obtain a p-value.

## Data and code sharing

The data analyzed in this study are publicly available on Figshare (*Kloosterman et al., 2019*). We programmed mMSE analysis in a MATLAB function within the format of the FieldTrip toolbox (*Oostenveld et al., 2011*). Our ft_entropyanalysis.m function takes as input data produced by Field-trip's ft_preprocessing.m function. In our function, we employed matrix computation of mMSE for increased speed, which is desirable due to the increased computational demand with multi-channel data analyzed with a sliding window. The function supports GPU functionality to further speed up computations. The software can be found on https://github.com/LNDG/mMSE. A tutorial for computing mMSE within the FieldTrip toolbox can be found on the FieldTrip website (http://www.field-triptoolbox.org/example/entropy_analysis/). Analysis scripts for the current paper can be found on https://github.com/kloosterman/critEEGentropy (*Kloosterman, 2020*; copy archived at https://github.com/elifesciences-publications/critEEGentropy/).

# Additional information

### Funding

| Funder | Grant reference number | Author |
|---|---|---|
| Max-Planck-Gesellschaft | Open-access funding | Niels A Kloosterman<br>Julian Q. Kosciessa<br>Ulman Lindenberger<br>Douglas D Garrett |
| Deutsche Forschungsge-meinschaft | Emmy Noether Programme grant | Niels A Kloosterman<br>Douglas D Garrett |

The funders had no role in study design, data collection and interpretation, or the decision to submit the work for publication.

### Author contributions

Niels A Kloosterman, Conceptualization, Data curation, Software, Formal analysis, Investigation, Visualization, Methodology, Writing - original draft, Project administration, Writing - review and editing; Julian Q Kosciessa, Formal analysis, Methodology, Writing - review and editing; Ulman Lindenberger, Supervision, Funding acquisition, Writing - review and editing; Johannes Jacobus Fahrenfort, Conceptualization, Resources, Data curation, Writing - review and editing; Douglas D Garrett, Conceptualization, Software, Supervision, Funding acquisition, Investigation, Project administration, Writing - review and editing

### Author ORCIDs

Niels A Kloosterman https://orcid.org/0000-0002-1134-7996
Julian Q Kosciessa https://orcid.org/0000-0002-4553-2794
Ulman Lindenberger http://orcid.org/0000-0001-8428-6453
Johannes Jacobus Fahrenfort http://orcid.org/0000-0002-9025-3436
Douglas D Garrett https://orcid.org/0000-0002-0629-7672

### Ethics

Human subjects: Human subjects: Participants provided written informed consent before the start of the experiment. All procedures were approved by the ethics committee of the Psychology Department of the University of Amsterdam (approval identifier: 2007-PN-69).

### Decision letter and Author response

Decision letter https://doi.org/10.7554/eLife.54201.sa1
Author response https://doi.org/10.7554/eLife.54201.sa2

## Additional files

**Supplementary files**
• Transparent reporting form

### Data availability

All data analyzed during this study are publicly available (https://doi.org/10.6084/m9.figshare.6142940.v1). Analysis scripts are publicly available on Github (https://github.com/kloosterman/critEEGentropy, copy archived at https://github.com/elifesciences-publications/critEEGentropy). A tutorial for computing mMSE within the FieldTrip toolbox (see https://github.com/LNDG/mMSE) has been published on the FieldTrip website (http://www.fieldtriptoolbox.org/example/entropy_analysis/).

The following previously published dataset was used:

| Author(s) | Year | Dataset title | Dataset URL | Database and Identifier |
|---|---|---|---|---|
| Kloosterman NA, de Gee JW, Werkle-Bergner M, Lindenberger U, Garrett DD, Fahrenfort JJ | 2018 | Humans strategically shift decision bias by flexibly adjusting sensory evidence accumulation in visual cortex | https://doi.org/10.6084/m9.figshare.6142940.v1 | figshare, 10.6084/m9.figshare.6142940 |

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
