## [Decision Letter]

Thank you for submitting your article "Boosting Brain Signal Variability Underlies Liberal Shifts in Decision Bias" for consideration by *eLife*. Your article has been reviewed by Michael Frank as the Senior Editor and Reviewing Editor, and three reviewers. The following individual involved in review of your submission has agreed to reveal their identity: Eelke Spaak (Reviewer #1).

The reviewers have discussed the reviews with one another and the Reviewing Editor has drafted this decision to help you prepare a revised submission.

Summary:

Kloosterman et al. conducted follow-up analyses of the data reported in their 2019 *eLife* paper. The experiment reported in both manuscripts required subjects to detect a relatively difficult perceptual target under instructions that either emphasized a liberal or a conservative decision bias, while recording EEG. In the previous paper the authors had reported that differences in oscillatory EEG activity in the alpha and the gamma range were associated with the different decision criteria. However, that study had not found any condition differences in the EEG signals that directly scaled with the individual differences in the liberal-conservative bias. The premise of the current study is that higher entropy in the brain response may underlie a regime that is more easily swayed in a particular direction and therefore can strategically implement a more liberal decision bias. Indeed, the authors report that within-subject, condition-specific differences in moment-to-moment EEG variability, as indexed by multi-scale entropy (mMSE), are strongly correlated with individual differences the decision-bias effect. These effects seemed located at frontal electrode sites and were observed across a relatively large span of time scales, and mainly after stimulus onset. The possibility that a liberal decision bias is implemented by upregulating neural noise is theoretically very interesting.

Essential revisions:

1) One of the conceptual issues with this paper is that no persuasive/convincing a priori theory given to relate these to concepts. More effort should be given to articulate the biological motivation to help readers appreciate the potential significance of the results. The authors interpret their results showing that "flexible adjustments of moment-to-moment variability in frontal regions may underlie strategic shifts in decision bias". However, the hallmark of a strategic effect is that the critical variable (i.e., entropy) actually differs on average between conditions that induce different strategies. However, inspection of figures suggests that liberal/conservative entropy condition differences were distributed rather evenly around zero (e.g., Figure 3B) – which is in contrast to the oscillatory effects in the previous paper that did show reliable condition differences. Thus, the results appear to suggest that while variability actually does affect the decision bias, it varies unsystematically across conditions within individuals. In other words, unless I am missing something it seems that the changes in entropy are *not* under top-down, strategic control. This aspect requires further discussion/analyses.

2) Power / sample size. In principle, the experimental design is adequate as it implements a robust within-subject bias manipulation (including substantial individual differences in this effect). The authors also develop a newly adapted, scale-free entropy measure that works within relatively short, concatenated time windows. The results seem surprisingly robust (a brain/behavior correlation of >.8) and are confirmed through within-subject block-by-block correlations (that were however considerably smaller). However, while the reported effects are highly significant, they also stem from analyses that seem relatively exploratory in nature (across all electrodes and time scales and time points), with a small sample of only 16 subjects. An independent replication would significantly strengthen the conclusions. Assuming my concern is valid and the main result is different than a-priori predicted (because it is not consistent with a strategic effect), the explanation comes across as more post-hoc, and as a consequence, the threshold for accepting a low-powered result would increase. The authors should therefore either replicate the experiment or provide strong arguments for why the concerns regarding robustness are unwarranted and/or why their results are consistent with a strategic effect after all.

To further unpack this statistical issue, with a relatively large search space (electrodes x time x timescale) plus multiple-testing correction only very high correlations can meet threshold. I find it difficult to determine whether or not this concern is real because I have a hard time fully understanding the cluster-correction procedure – in particular over which electrodes data were averaged for the scatterplot. In the topographic images there are some electrodes sites that seem to show significant correlations based on the coloring scheme, but there are also varying numbers of black circles (sometimes of varying size). Are results averaged cross colored electrodes, or "circled" electrodes? What is the significance of the circle size? The broader the base of averaging, the less I would be the concerned. I find the split-half robustness check with such a high correlation to begin with not extremely re-assuring. However, the fact that the relationship is also found within subjects (though much smaller) does clearly help.

3) All reviewers felt that the methods by which the mMSRE is computed should be better articulated, particularly given that the manuscript uses a modified version of a published technique. All details should be made explicit. To name just one example, in subsection “Entropy computation”: "estimation of SampEn involves counting… (p^m)…" Many of these statements are ambiguous – instead a step-by-step description of the algorithm should be given. Equations would also help to make the Materials and methods section more readable.

One reviewer also felt that it might help to restructure the Results section such that quantities closer to the measured data are presented first, before building on those to get to the change-change correlation. For example, the section should start with line graphs of mMSE over trial time per condition, etc. That way, readers can appreciate much better what is going on.

4) Reviewers also all agreed that the analyses should be repeated without first removing the ERPs. The authors subtract the condition-, session-, and participant-wise ERP from the raw trials before all analyses. While this is sometimes warranted, if the individual trial responses are not stereotypical, the subtraction procedure can actually *add* the (inverted) ERP back into the data. More specifically, ERP subtraction to obtain induced-only data is only justified if one can assume that the (single-trial) evoked response is independent from (i.e., adds linearly to) the "ongoing" activity. In the present study, the authors are specifically concerned with how ongoing activity influences the processing of new stimuli, thus by definition violating this assumption. Therefore, in this case, the subtraction of the ERP is problematic and should be avoided. Relatedly, another reviewer noted that one potential contributing factor to the results is that trials where a stronger (e.g. temporal) deviation of the EEG signal from the average ERP leads to lower entropy, since there the 'evoked' response is less efficiently removed, whereas in trials where the EEG resembles the ERP more closely, entropy will be higher since the stereotypical response has been removed. Thus, please repeat the analysis without removing the ERP and also show the ERPs to the different conditions.

5) The text in Results section suggests that the spectral analysis has been conducted on both baseline-corrected spectral values as well as raw power (but this is not stated in the methods section where it only mentions baseline corrected power values). Since the experimental design is blocked, looking at non-baseline corrected values is essential. I would like the authors to provide an analysis of spectral power (across frequency ranges) which is not baseline corrected, but where possibly the average power of each participant and electrode across conditions have been subtracted – to account for inter-individual differences in EEG signal power that may be unrelated to decision processes (conductivity, skull thickness, dipolar orientation etc).

6) The authors highlight that it might not be the theta-amplitude but other characteristics of the theta band oscillation that contribute to the findings. This would probably be best investigated by providing an analysis on bandpass-filtered data (ideally approximately within a broader range that encompasses the band where amplitude differences have been found within conditions – and compare these to other bands or the non-bandpass filtered signal.

[Editors' note: further revisions were suggested prior to acceptance, as described below.]

Thank you for resubmitting your article "Boosting Brain Signal Variability Underlies Liberal Shifts in Decision Bias" for consideration by *eLife*. Your revised article has been reviewed by Michael Frank as the Senior Editor and Reviewing Editor, and two reviewers. The following individual involved in review of your submission has agreed to reveal their identity: Eelke Spaak (Reviewer #1).

The reviewers have discussed the reviews with one another and the Reviewing Editor has drafted this decision to help you prepare a revised submission.

Both reviewers found this version to be an improvement over the first iteration. The theoretical embedding in Introduction and Discussion is more convincing, and the clarity of the Methods is much improved. They were also happy to see that the results remain unaltered when ERPs are vs. are not subtracted, and the new addition of the high-pass filtering analysis gives a welcome piece of extra insight into what the mMSE metric is picking up on. The inclusion of ERP-based analyses is useful, as is the demonstration that filtering of lower frequencies modulates the observed relationships.

However, the following issues remained and should be addressed.

1) The reviewers were more or less convinced by the authors' narrative that across-participant correlations pick up on something different than condition-wise main effects. However, they both felt that it may be overreaching to interpret this effect as

"strategic" – which would really have required a main effect of the conservative vs. liberal condition. Both reviewers noted that the main points of the paper do not hinge on this being a true strategic, top-down controlled, effect, though, and hence we would like you to tone down these claims.

In more detail, points from each of the reviewers:

"The authors' argument: "certain participants may have used the correct and others the incorrect strategy" (Discussion section) is unsatisfactory. If that were the case, then the subgroup of participants with high behavioural conservative vs liberal criterion difference should show a stronger brain/behaviour change/change correlation than the subgroup of participants with a low behavioural con vs lib difference. That is a different prediction (i.e. an interaction between behavioural difference (group) and change/change correlation) than the existence of a significant change/change correlation per se. Based on Figure 3B, it's clear that such an interaction is not present in the data. Note that I do *not* suggest the authors actually test for this interaction! It's extremely derivative and would require far more participants than present here. I'm mentioning it to show that the authors' argument why the correlation can be a consequence of a strategic effect is invalid. Instead, the authors should probably concede that the effects reported here are likely non-strategic."

and

"I appreciate the full discussion of the mean vs. variability situation. I fully agree that correlations and mean effects provide orthogonal information and that it is logically possible that one half of the participants do strategically the right thing and the other to strategically the wrong thing, resulting in robust change-change effects in the absence of mean differences. My point is simply that it strikes me as rather implausible that our brain provides a strategic mechanism for such a fundamental problem as criterion setting that only half of the people are able to utilize in an adaptive manner. I wonder whether an alternative interpretation is that the variability waxes and wanes endogenously and/or differs across people in an uncontrollable manner, and that it is only through an additional, strategic process (e.g., indexed through the theta activity) that this variability is gated towards affecting the threshold setting. In other words, everybody knows when to reach for the hammer when needing to hit a nail, but the quality of the hammer varies between (and within) people. (Maybe this is even something that could be tested in the within subject analyses, by looking at the interaction between theta boosts and entropy in affecting behavior.)"

2) One of the reviewers noted that they still want to see "raw" mMSE values. The authors' reply ("plotting mMSE is tricky because it has time/space/timescale dimensions") is very unconvincing, since in Figure 3A they are showing change/change correlation values exactly for those three dimensions as well. So: the authors should simply please show two plots, exactly as in Figure 3A, one for Liberal, one for Conservative, while showing raw mMSE (condition- and subject-wise averaged) values, rather than change/change correlations. The channels picked for the time/timescale colour plot should be identical to those channels that come out of the change/change correlation cluster test, and the time/timescale voxels for the topoplot should be identical to those picked for that test as well (so basically use the same data selection as in Figure 3A; no need to separately test mMSE against zero or anything). Adding these plots (if the authors prefer: as a supplement) is critical for readers to appreciate what aspects of the signal mMSE (and the derived metrics of mMSE-change and its correlation with behaviour) is picking up on, and thus to allow them to evaluate the importance of the conclusions here.

3) It was suggested to show correlations in some condensed manner separately for liberal and conservative conditions (aside form mean values). The difference score sort of assumes that both conditions contribute, but that need not be the case. For example, it is at least possible that the relationship is produced through the liberal condition only, whereas the effect of variability might be squelched in the conservative condition (which actually would strengthen the strategy argument). Not likely, but worth exploring.

---

## [Author Response]

Essential revisions:1) One of the conceptual issues with this paper is that no persuasive/convincing a priori theory given to relate these to concepts. More effort should be given to articulate the biological motivation to help readers appreciate the potential significance of the results. The authors interpret their results showing that "flexible adjustments of moment-to-moment variability in frontal regions may underlie strategic shifts in decision bias". However, the hallmark of a strategic effect is that the critical variable (i.e., entropy) actually differs on average between conditions that induce different strategies. However, inspection of figures suggests that liberal/conservative entropy condition differences were distributed rather evenly around zero (e.g., Figure 3B) – which is in contrast to the oscillatory effects in the previous paper that did show reliable condition differences. Thus, the results appear to suggest that while variability actually does affect the decision bias, it varies unsystematically across conditions within individuals. In other words, unless I am missing something it seems that the changes in entropy are NOT under top-down, strategic control. This aspect requires further discussion/analyses.

We thank the reviewers for these points. We agree that our central hypothesis that increased moment-to-moment neural variability underlies a liberal decision bias should be articulated further and better embedded in a theoretical framework. Our fundamental hypothesis is that a neural system that is more dynamic and less predictable allows cognitive functioning to be more flexible and adaptive. We have now specified the theoretical underpinnings of our hypothesis better in the Introduction, based on three independent lines of research. First, we draw on evidence from studies showing that increased neural variability allows the brain to continuously explore possible internal states, which may help to respond flexibly whenever a task-relevant event occurs. Evidence for this idea comes both from resting-state connectivity work showing that neural noise is essential for the emergence of the coherent fluctuations of the default network (Ghosh et al., 2008), as well as from task data showing that more meaningful stimuli elicit higher levels of neural variability, suggesting more elaborate processing (Mišić et al., 2010). This notion is further reinforced by several papers from our lab showing that higher neural variability is associated with better cognitive performance (Garrett et al., 2015, 2013, 2011). Second, we draw from theoretical work suggesting that organisms can employ a high-fidelity neural regime that is able to mirror the complexity of the environment and thus increases chances of detecting relevant stimuli, but at the cost of higher energy demand than in a low-fidelity regime (Marzen and DeDeo, 2017). High-fidelity encoding has been proposed to be reflected in increased neural variability (Młynarski and Hermundstad, 2018), and we have recently provided initial empirical evidence for this idea (Garrett et al., 2020). Finally, we draw from influential dynamical attractor models of neural population activity, in which temporal neural variability (operationalized as “noise” in these models) enables probabilistic jumping across barriers within a high-dimensional energy landscape to allow settling into a low-dimensional attractor state that corresponds to a particular cognitive outcome (Deco et al., 2009; Moreno-Bote et al., 2007; Wimmer et al., 2014). Together, these lines of work suggest that neural variability supports cognitive function and flexibility, which we seek to reinforce and extend with the current study in the domain of decision biases.

The reviewers correctly state that on average (across participants), raw mMSE in frontal electrodes is not significantly higher in the liberal compared to the conservative condition (indicating no main effect of condition). This is in contrast to raw alpha and theta power differences, which significantly differed between conditions (see our previous paper). Paradoxically, however, we found that whereas the shift in mMSE did correlate with the liberal-conservative bias shift, the shifts in alpha and theta did not; this is indeed the “research advance” we provide with the current paper. We argue that an empirical dissociation of main effects and behavioral prediction is not unexpected. Although a main effect of experimental condition and a change-change correlation can in principle occur in parallel, inter-individual differences may be sensitive to behavioral adjustments even in the absence of a group-wise shift. The reason for this is that a main effect and a correlation address somewhat opposite research questions that can be in conflict with each other. On the one hand, a main effect tends to occur when subjects respond similarly to an experimental condition, which typically necessitates that individual differences are relatively small. On the other hand, chances of detecting a correlation with e.g., behavior typically increase when individual differences are larger (e.g., Lindenberger et al., 2008). In short, the presumption that a main effect of condition is a prerequisite for detecting behavioral effects is unfounded in our view. Further, the idea that evidence for top-down “strategies” requires a main effect of condition is also not warranted; certain subjects may use the correct and others the incorrect strategy, potentially leading to a lack of average across-subject condition effect, yet revealing why their behaviors differed on task. Therefore, we were not seeking to a priori hypothesize a main effect of condition in the liberal-conservative contrast, but rather focused our hypotheses on inter-individual adjustments in neural entropy that tracked the magnitude of the individual bias shift. The observed lack of a main effect combined with a strong correlation with the bias shift indicates that some participants indeed showed slightly higher mMSE in the conservative condition, but that these participants also showed the smallest bias shift between conditions. This suggests that these participants responded only weakly both on the neural and behavioral levels. Therefore, we argue that the changes in mMSE still reflect a strategic bias shift, even in the absence of a main effect. Finally, we note that we do indeed observe a significantly stronger transient mMSE increase after trial onset in the liberal condition (after baseline-correcting the data, as is common in time-frequency EEG analysis). Perhaps mediated by the logic outlined above, the change-change correlation with behavior is weaker in this version. We now discuss this issue regarding the link between main effect of condition and change-change correlation in the Discussion section of the manuscript.

2) Power / sample size. In principle, the experimental design is adequate as it implements a robust within-subject bias manipulation (including substantial individual differences in this effect). The authors also develop a newly adapted, scale-free entropy measure that works within relatively short, concatinated time windows. The results seem surprisingly robust (a brain/behavior correlation of >.8) and are confirmed through within-subject block-by-block correlations (that were however considerably smaller). However, while the reported effects are highly significant, they also stem from analyses that seem relatively exploratory in nature (across all electrodes and time scales and time points), with a small sample of only 16 subjects. An independent replication would significantly strengthen the conclusions. Assuming my concern is valid and the main result is different than a priori predicted (because it is not consistent with a strategic effect), the explanation comes across as more post-hoc, and as a consequence, the threshold for accepting a low-powered result would increase. The authors should therefore either replicate the experiment or provide strong arguments for why the concerns regarding robustness are unwarranted and/or why their results are consistent with a strategic effect after all.

Thank you for these valid points. In response to point 1 above, we have already argued that the lack of a main effect does not preclude the interpretation that interindividual differences in the magnitude of entropy shifts reflect strategic bias adjustments. For example, some participants have slightly higher entropy in the conservative condition, but their strategic bias shift was also small, hence eliminating a main entropy effect while retaining strong coupling in their changes. As noted in response to point 1, we have now motivated our hypothesis better and argue why we did not hypothesize a main effect of condition per se. Moreover, when baseline-correcting mMSE values, as routinely done in time-frequency analysis, we indeed observe a significant main effect of the bias manipulation with higher entropy in the liberal condition (Figure 6E), but a weaker change-change correlation with behavior. We chose to test the change-change relationship without first applying a pre-stimulus baseline correction of the mMSE values, because our block-wise design with continuous stimulation presumes the strategic bias also to be present before stimulus onset, and the mMSE computation already involves two within-subject normalizations, in contrast to raw spectral power, which is not grounded in a subject’s own brain activity in any way. Specifically, such grounding is ensured for mMSE by (1) the pattern similarity parameter controlling for the subject’s overall signal variation (SD), while (2) the division of m and m+1 pattern counts in the entropy computation normalizes the absolute counts of m and m+1 pattern matches. However, we also included the baseline-corrected results in our manuscript to investigate possible mMSE quenching effects and allow the comparison with the spectral power modulation results.

We were initially also surprised by the strong brain-behavior correlation, especially because none of the spectral or ERP measures shows any convincing, significant correlation. As noted in the manuscript, some correlation would be expected if these measures reflect a neural process central to the bias shift, given the large amounts of data per participant and the large inter-individual behavioral differences. Additionally, we found that the correlation was also significant for more spontaneous bias fluctuations within participants, suggesting that the observed link between mMSE and decision bias not only occurs under strategic control, but might also be subject to moment-to-moment fluctuations in attention or vigilance. In the end, the many controls (including ERPs and the new high-pass filtering analysis included in this revision) convinced us that entropy uniquely captures some part of the neural dynamics that more conventional measures do not.

We argue that a number of specifics of our cluster-based correlation analysis speak against the idea of overfitting a higher-dimensional spatiotemporal space. First, the spatial topography of our mMSE effect is highly similar to midfrontal theta, and even correlates with theta, suggesting overlapping neural sources. Second, the analysis shows that many time scales contribute to the correlation, suggesting that low frequencies such as delta/theta contribute to the correlation, given the progressive low-pass filtering with longer time scales (see also our new analysis reported under reviewer point 6). Previous work indeed has implicated delta/theta at frontal channels in strategic and cognitive control of decision making (Helfrich et al., 2017). Third, the correlation extends widely in the time dimension, suggesting a stable strategic bias shift over time, as was required of the participants in our block-wise design. Finally, the correlation analysis remains virtually unchanged independent of whether the event-related potential is removed before entropy analysis (see our response to reviewer point 4 for details), suggesting robustness (Mišić et al., (2010) found a similar result). Taken together, we feel that despite the unexpectedly strong brain-behavior link for mMSE, the results are robust and consistent with the literature in several dimensions. We have highlighted these various aspects better in the manuscript to communicate this more clearly. On a final note, we agree that our sample size, in terms of number of participants, is relatively small and that a replication of our findings in an independent sample of participants is highly desirable. Nevertheless, we think that the present results sufficiently strong and conceptually motivated to warrant publication on their own, given the massive amount of data within subjects and robustness against the many controls.

To further unpack this statistical issue, with a relatively large search space (electrodes x time x timescale) plus multiple-testing correction only very high correlations can meet threshold. I find it difficult to determine whether or not this concern is real because I have a hard time fully understanding the cluster-correction procedure-in particular over which electrodes data were averaged for the scatterplot. In the topographic images there are some electrodes sites that seem to show significant correlations based on the coloring scheme, but there are also varying numbers of black circles (sometimes of varying size). Are results averaged cross colored electrodes, or "circled" electrodes? What is the significance of the circle size? The broader the base of averaging, the less I would be the concerned. I find the split-half robustness check with such a high correlation to begin with not extremely re-assuring. However, the fact that the relationship is also found within subjects (though much smaller) does clearly help.

This comment made us aware that the way in which we plot the results was not entirely clear, so we have tried to improve this. For the correlation analysis, we used a cluster-based permutation test across the time, electrode and timescale dimensions. This analysis is highly similar to the commonly applied statistical testing of E/MEG spectral power (modulation) across the time, electrodes and frequency dimensions. To this end, we used the cluster-based statistics functionality in the ft_freqstatistics.m function (Maris and Oostenveld, 2007) as implemented in the FieldTrip toolbox (Oostenveld et al., 2011). As the reviewer points out, this method is most sensitive to strong correlations that extend widely across the three dimensions. However, we chose not to select an a priori region or time/timescale range of interest for investigating the correlation because we were interested whether there would be any region that would show an effect. Indeed, we found a strong, significant correlation for mMSE, but not spectral power, and now also not for ERPs (see reviewer point 4). Thus, despite being sensitive mostly to strong, extended clusters of correlations, the method allows us to reveal an effect for mMSE over and above other, conventional measures of neural activity without having to preselect timescales, timepoints and electrodes of interest.

Importantly, the scatter plot depicted in Figure 3B serves only to illustrate the cluster-based correlation test reported in Figure 3A, and bears no additional information over the statistical test reported in Figure 3A. We obtained this scatter plot by first averaging the liberal-conservative mMSE values across all the time-timescale-electrode bins that were within the significant three-dimensional cluster depicted in Figure 3A, and correlating with the liberal-conservative SDT criterion across participants. Given that the bins were preselected based on their correlation in the first place and then averaged, the reported correlation of rho = 0.87 represents an upper bound of the actual correlation. Electrodes included in the cluster are shown via black dots, and values were extracted exclusively within these significant channels. Since the coloring of the electrode locations is done through interpolation, it can occur that the indicated electrode locations do not exactly match the colors in the topoplots. Also, it can occur that no color is present at an electrode location – this indicates that although the electrode was represented in the cluster, its contribution was so small that it did not go outside the gray range of the color bar. We have now clarified these illustration principles in the legend of Figure 3A.

3) All reviewers felt that the methods by which the mMSRE is computed should be better articulated, particularly given that the manuscript uses a modified version of a published technique. All details should be made explicit. To name just one example, in subsection “Entropy computation”: "estimation of SampEn involves counting… (p^m)…" Many of these statements are ambiguous – instead a step-by-step description of the algorithm should be given. Equations would also help to make the Materials and methods section more readable.

We agree that our explanation of our modified MSE measure could be improved, so we have thoroughly revised the Materials and methods section with this point in mind. We now describe step by step how MSE is computed, and also provide equations that convey the logic of the computation. We hope that the reviewers find the entropy computation and our modifications now easier to understand. We now write the following in subsection “Entropy computation”:

**“**We measured temporal neural variability in the EEG using multiscale entropy (MSE)(Costa et al., 2002). MSE characterizes signal irregularity at multiple time scales by estimating sample entropy (SampEn) of a signal’s time series at various sampling rates. The estimation of SampEn involves counting how often specific temporal patterns reoccur over time, and effectively measures how unpredictably the signal develops from moment to moment. At a given time scale, the estimation of SampEn consists of the following steps:[…]

Thus, SampEn estimates the proportion of similar sequences of m samples that are still similar when the next sample, i.e., m+1, is added to the sequence. Here, we use m=2 and r=0.5, as typically done in neurophysiological settings (Courtiol et al., 2016; Grandy et al., 2016; Richman and Moorman, 2000).”

We now also separately highlight our adaptations to this formula for clarity and comparability with previous implementations, please see Materials and methods section.

One reviewer also felt that it might help to restructure the Results section such that quantities closer to the measured data are presented first, before building on those to get to the change-change correlation. For example, the section should start with line graphs of mMSE over trial time per condition, etc. That way, readers can appreciate much better what is going on.

This is an interesting suggestion that we have given much thought while writing the manuscript. Since the mMSE output has three dimensions (time, space, and timescales), one issue with plotting one-dimensional time courses of mMSE as line graphs is that this requires an arbitrary choice of electrodes and timescales to average over before plotting. (A similar problem arises for the presentation of three-dimensional values of spectral power in time, space and frequency.) In addition, we do not first convert the mMSE into a percentage modulation from the pre-stimulus baseline, as is routinely done for spectral estimates, which prohibits a statistical test against zero that could help to select relevant electrodes and timescales. Please see our response to reviewer point 2 for why we chose to primarily work with raw, non-baseline-corrected mMSE values. Therefore, we reasoned that it might be misleading to show line graphs of mMSE and decided to go straight to the change-change correlation with behavior across all dimensions.

4) Reviewers also all agreed that the analyses should be repeated without first removing the ERPs. The authors subtract the condition-, session-, and participant-wise ERP from the raw trials before all analyses. While this is sometimes warranted, if the individual trial responses are not stereotypical, the subtraction procedure can actually add the (inverted) ERP back into the data. More specifically, ERP subtraction to obtain induced-only data is only justified if one can assume that the (single-trial) evoked response is independent from (i.e., adds linearly to) the "ongoing" activity. In the present study, the authors are specifically concerned with how ongoing activity influences the processing of new stimuli, thus by definition violating this assumption. Therefore, in this case, the subtraction of the ERP is problematic and should be avoided. Relatedly, another reviewer noted that one potential contributing factor to the results is that trials where a stronger (e.g. temporal) deviation of the EEG signal from the average ERP leads to lower entropy, since there the 'evoked' response is less efficiently removed, whereas in trials where the EEG resembles the ERP more closely, entropy will be higher since the stereotypical response has been removed. Thus, please repeat the analysis without removing the ERP and also show the ERPs to the different conditions.

We have given the relationship between the ERP and entropy much thought during this project, and decided to remove the event-related potentials (ERPs) in our initial submission for two reasons. First, we aimed to focus our analyses on neural activity that originates intrinsically in the brain, rather than being directly evoked by sensory stimulation. This was also motivated by our block-wise design which required our participants to keep a strategic bias online continuously. In addition, ERP removal also removed the strong steady-state visual evoked potential at 25 Hz in occipital electrodes that we reported in our previous paper (Kloosterman et al., 2019), which could affect the entropy estimates at faster scales. Second, when presenting the results to different audiences and during internal rounds of manuscript review we were often asked whether the entropy results could be explained by ERP differences between conditions, since computing ERPs is a well-known and comparably simpler analysis than entropy analysis. To address these issues, we decided to subtract the ERP before computing entropy in our initial submission. However, we agree with the reviewers that besides removing evoked responses, subtracting the ERP can have unwanted consequences such as introducing a sign-flipped ERP back into the data. We therefore repeated the analysis without removing the ERP and found very similar results, with a virtually unchanged negative change-change correlation between bias and entropy. This is in line with a previous paper that also showed similar MSE results with and without ERF removal in MEG data (Mišić et al., 2010). We now report the ERPs for the two conditions in Figure 3—figure supplement 4, and report their correlation with behavior. In the liberal-conservative contrast, we observed one positive cluster (i.e. stronger ERP for liberal) over central and parietal electrodes between 0.37 s and 0.8 s after trial onset, and one negative cluster in midfrontal, central and parietal electrodes slightly earlier in time. The timing and topography of the former, positive cluster closely corresponds to an ERP known as the centroparietal positivity (CPP), which reflects sensory evidence accumulation during perceptual decision making (O’Connell et al., 2012). This is consistent with our previous finding of increased evidence accumulation in the liberal condition (Kloosterman et al., 2019). Importantly for the current paper, we also performed the change-change correlation between ERPs and criterion across electrodes and time bins, and found no significant clusters (lowest cluster p-value p = 0.69 for positive cluster, and p = 0.23 for negative cluster). Taken together, these results suggest that ERPs time-locked to stimulus onset cannot account for the observed entropy results.

On a final note, we would like to point out that this result intriguingly suggests that the large field of ERP research (as perhaps the most widely used EEG analysis method) might have structurally overlooked a crucial property of the EEG signal that is in fact strongly linked to behavior, and that this link has remained hidden in many existing ERP-analyzed EEG datasets. We discuss these results and their significance in the manuscript in a new subsection “Entropy-bias relationship is not explained by event-related potentials*”* and in the Discussion section.

5) The text in Results section suggests that the spectral analysis has been conducted on both baseline-corrected spectral values as well as raw power (but this is not stated in the methods section where it only mentions baseline corrected power values). Since the experimental design is blocked, looking at non-baseline corrected values is essential. I would like the authors to provide an analysis of spectral power (across frequency ranges) which is not baseline corrected, but where possibly the average power of each participant and electrode across conditions have been subtracted – to account for inter-individual differences in EEG signal power that may be unrelated to decision processes (conductivity, skull thickness, dipolar orientation etc)

Following the reviewers’ recommendations, we performed a baseline correction on the raw spectral power estimates by subtracting within each subject the across-condition average power in the pre-stimulus period (–0.4 to 0 s) from each time-frequency bin in each electrode. Please see the new Figure 3—figure supplement 3 for the results. Overall, the modulations using this baseline closely resemble the power modulations using a condition-specific baseline as reported in our previous paper (Kloosterman et al., 2019), including the SSVEP responses over posterior regions, gamma enhancement in occipital electrodes, and low-frequency suppression in central and occipital electrodes (Figure 3—figure supplement 3A and B). Contrasting the two conditions revealed stronger overall high frequency power as well as suppressed alpha-band activity for the liberal condition (panel C). Importantly, however, when change-change correlating liberal-conservative power modulation with the decision bias shift, we found no significant clusters (lowest cluster p = 0.26, Figure 3—figure supplement 3D). Thus, this specific baseline correction also did not reveal any across-participant link between spectral power modulation and decision bias, in line with a unique contribution of entropy shifts to the bias shift. Please note that since we now report non-ERP-removed results, we removed the prior analysis (i.e., spectral analysis on data with ERP-removed and per-condition baseline-corrected) from the current revised manuscript, but still report the lack of across-participant correlation between per-condition baseline corrected modulation and the bias shift in the manuscript. We refer to our previous paper for detailed spectral analysis with this per-condition baseline without ERP removal (Kloosterman et al., 2019).

6) The authors highlight that it might not be the theta-amplitude but other characteristics of the theta band oscillation that contribute to the findings. This would probably be best investigated by providing an analysis on bandpass-filtered data (ideally approximately within a broader range that encompasses the band where amplitude differences have been found within conditions – and compare these to other bands or the non-bandpass filtered signal.

An intriguing question indeed remains exactly which characteristic of the time-domain EEG signal underlies the link between mMSE and decision bias. Since we previously found stronger oscillatory activity in the delta/theta frequency ranges (2-6 Hz) in frontal electrodes (Kloosterman et al., 2019), we examined the impact of systematically removing the lowest frequencies from the data on the strength of the observed brain-behavior relationship. This reflects a stringent test for the necessity of narrowband frequency ranges for the brain-behavior relationship. To this end, we applied a Butterworth high-pass filter with different cutoff frequencies to the time-domain data (filter order of 4, trials mirror-padded to 4 s to allow robust estimation (Cohen, 2014), and subsequently applied mMSE analysis. To quantify the strength of the brain-behavior correlation at each filtering step and allow unbiased comparisons across cutoff frequencies, we finally averaged mMSE within the time-space-timescale cluster that showed the strong negative correlation in our principal (non-filtered) analysis (see Figure 3B), and plotted scatterplots of the correlation. Figure 4 shows the results of this analysis for non-filtered (original result, panel A), 1, 2, and 3 Hz high-pass filters (panels B-D). (Please note that we applied a 0.5 Hz high-pass filter on continuous data during preprocessing to remove drifts in all variants.) Interestingly, we observe that the brain-behavior relationship progressively weakens as the cutoff frequency increases, such that the correlation is non-significant (but still negative) after applying a 3 Hz high-pass filter prior to entropy estimation.

Whereas this finding suggests that these low frequencies are necessary for the mMSE-bias relationship, our previous control analyses reported in the initial submission indicate that statistically controlling for 1-2 Hz (delta) power does not affect the brain-behavior relationship (Figure 3—figure supplement 3G). One explanation for these seemingly incongruent results could be the different ways in which oscillatory phase is treated in these two analyses: whereas statistically controlling for delta power fully ignores phase, the high-pass filtering analysis removes both power and phase from the signal before entropy is computed. Moreover, the power control disregards potential interactions between 1-2 Hz and higher frequencies (e.g. theta) that could contribute to the relationship. Taken together, these analyses reveal that the lowest frequencies present in the data might play a role in the entropy-behavior relationship, possibly through non-linear interactions between spectral power and spectral phase of these frequencies. We have added this analysis to the manuscript and propose to address the exact non-linear relationships between entropy, spectral power and phase in future work.

[Editors' note: further revisions were suggested prior to acceptance, as described below.]

However, the following issues remained and should be addressed:1) The reviewers were more or less convinced by the authors' narrative that across-participant correlations pick up on something different than condition-wise main effects. However, they both felt that it may be overreaching to interpret this effect as"strategic" – which would really have required a main effect of the conservative vs liberal condition. Both reviewers noted that the main points of the paper do not hinge on this being a true strategic, top-down controlled, effect, though, and hence we would like you to tone down these claims.In more detail, points from each of the reviewers:"The authors' argument: "certain participants may have used the correct and others the incorrect strategy" (Discussion section) is unsatisfactory. If that were the case, then the subgroup of participants with high behavioural conservative vs liberal criterion difference should show a stronger brain/behaviour change/change correlation than the subgroup of participants with a low behavioural con vs lib difference. That is a different prediction (i.e. an interaction between behavioural difference (group) and change/change correlation) than the existence of a significant change/change correlation per se. Based on Figure 3B, it's clear that such an interaction is not present in the data. Note that I do *not* suggest the authors actually test for this interaction! It's extremely derivative and would require far more participants than present here. I'm mentioning it to show that the authors' argument why the correlation can be a consequence of a strategic effect is invalid. Instead, the authors should probably concede that the effects reported here are likely non-strategic."and"I appreciate the full discussion of the mean vs. variability situation. I fully agree that correlations and mean effects provide orthogonal information and that it is logically possible that one half of the participants do strategically the right thing and the other to strategically the wrong thing, resulting in robust change-change effects in the absence of mean differences. My point is simply that it strikes me as rather implausible that our brain provides a strategic mechanism for such a fundamental problem as criterion setting that only half of the people are able to utilize in an adaptive manner. I wonder whether an alternative interpretation is that the variability waxes and wanes endogenously and/or differs across people in an uncontrollable manner, and that it is only through an additional, strategic process (e.g., indexed through the theta activity) that this variability is gated towards affecting the threshold setting. In other words, everybody knows when to reach for the hammer when needing to hit a nail, but the quality of the hammer varies between (and within) people. (Maybe this is even something that could be tested in the within subject analyses, by looking at the interaction between theta boosts and entropy in affecting behavior.)"

We appreciate the reviewers’ arguments questioning our previous conclusion that a strategic modulation of neural variability underlies a strategic decision bias shift. Although we maintain that a main effect of neural variability does not speak to whether an individual may have been more or less strategic than another (which are individual differences dependent), we agree that both our general conclusion of a strategic origin of neural variability shifts, as well as of their causal role in the behavioral bias adjustment may have been unwarranted given the present data. For this reason, we have chosen to stay closer to the data and now interpret the results in light of the adaptive cognitive process that the experimental bias manipulations required of the subjects in order to avoid penalties. This relaxes the assumption that these neural shifts arose from a strategic process and allows for non-deliberative factors, while retaining the central notion supported by the experimental design that these shifts are behaviorally adaptive. Although we feel it is indeed highly plausible that subjects may have deliberately and strategically implemented a particular mindset while doing the task (as also suggested by the frontal topography of the correlation), estimating this in detail would require additional experiments to verify, involving e.g. verbal reports of the strategy subjects used to solve the task. To reflect this amended stance, our interpretation in the Discussion section now reads as follows:

“Here, we provide first evidence that greater bias shifts are typified in those subjects who exhibit greater shifts in frontal mMSE after stimulus onset, suggesting that mMSE provides a core signature of such adaptive behavioral shifts.”

Finally, we now refrain from drawing direct links between mMSE and strategy throughout the manuscript, and have changed the manuscript title to better reflect the correlational nature of our results (mMSE “tracks” bias shift instead of “underlies”). We hope this appropriately addresses the reviewers’ concerns that the neural variability shifts might not have been strategic.

2) One of the reviewers noted that they still want to see "raw" mMSE values. The authors' reply ("plotting mMSE is tricky because it has time/space/timescale dimensions") is very unconvincing, since in Figure 3A they are showing change/change correlation values exactly for those three dimensions as well. So: the authors should simply please show two plots, exactly as in Figure 3A, one for Liberal, one for Conservative, while showing raw mMSE (condition- and subject-wise averaged) values, rather than change/change correlations. The channels picked for the time/timescale colour plot should be identical to those channels that come out of the change/change correlation cluster test, and the time/timescale voxels for the topoplot should be identical to those picked for that test as well (so basically use the same data selection as in Figure 3A; no need to separately test mMSE against zero or anything). Adding these plots (if the authors prefer: as a supplement) is critical for readers to appreciate what aspects of the signal mMSE (and the derived metrics of mMSE-change and its correlation with behaviour) is picking up on, and thus to allow them to evaluate the importance of the conclusions here.

We agree with the reviewer that it is helpful to have some intuition of the actual mMSE values in the two conditions to appreciate the change-change behavioral correlation. Therefore, we have followed the reviewers request and now report subject-averaged topoplots and time-timescale colour plots of mMSE within the significant correlation cluster identified in Figure 3A, separately for the two conditions (see Figure 3—figure supplement 1). Indeed, we see that on the subject-averaged level, mMSE for the two conditions is highly similar, in line with our previously reported lack of a main effect of condition. Subtracting the two conditions indeed shows only small, non-significant differences between the conditions, (see Figure 3—figure supplement 1, bottom panels). This figure again highlights the dissociation between main effects and correlation in our data: while we see no main effect of condition, we do observe a strong across-participant correlation between liberal-conservative shifts in mMSE and criterion (Figure 3).

3) It was suggested to show correlations in some condensed manner separately for liberal and conservative conditions (aside form mean values). The difference score sort of assumes that both conditions contribute, but that need not be the case. For example, it is at least possible that the relationship is produced through the liberal condition only, whereas the effect of variability might be squelched in the conservative condition (which actually would strengthen the strategy argument). Not likely, but worth exploring.

To explore this possibility, we repeated the correlation separately for the conservative and liberal conditions and found weak, non-significant, negative correlations in both (conservative: rho = –0.12, p = 0.66, liberal: rho = –0.21, p = 0.43). Moreover, we found no significant difference in correlation strength between the conditions (liberal-conservative ∆rho = –0.09, p = 0.7, non-parametric correlation difference test, 10.000 permutations), suggesting that the correlation is not driven by only one of the two conditions. Rather, this finding suggests that focusing on the bias shift by taking the within-subject subtraction is indeed necessary to bring out the link with behavior, and that both conditions contribute. We now report these results in subsection “The entropy-bias relationship is not explained by total signal variation or spectral power”.